# Bullying10K: A Large-Scale Neuromorphic Dataset towards Privacy-Preserving Bullying Recognition

**Yiting Dong**[1,3,4]*, **Yang Li**[2,3,4]*, **Dongcheng Zhao**[3,4]*, **Guobin Shen**[1,3,4], **Yi Zeng**[1,2,3,4]†

[1] School of Future Technology, University of Chinese Academy of Sciences
[2] School of Artificial Intelligence, University of Chinese Academy of Sciences
[3] Brain-inspired Cognitive Intelligence Lab,
Institute of Automation, Chinese Academy of Sciences
[4]Center for Long-term Artificial Intelligence

## Abstract

The prevalence of violence in daily life poses significant threats to individuals' physical and mental well-being. Using surveillance cameras in public spaces has proven effective in proactively deterring and preventing such incidents. However, concerns regarding privacy invasion have emerged due to their widespread deployment. To address the problem, we leverage Dynamic Vision Sensors (DVS) camera to detect violent incidents and preserve privacy since it captures pixel brightness variations instead of static imagery. We introduce the Bullying10K dataset, encompassing various actions, complex movements, and occlusions from real-life scenarios. It provides three benchmarks for evaluating different tasks: action recognition, temporal action localization, and pose estimation. With 10,000 event segments, totaling 12 billion events and 255 GB of data, Bullying10K contributes significantly by balancing violence detection and personal privacy persevering. And it also poses a challenge to the neuromorphic dataset. It will serve as a valuable resource for training and developing privacy-protecting video systems. The Bullying10K opens new possibilities for innovative approaches in these domains.

## 1 Introduction

The issue of violence in daily life poses a significant threat to individuals' physical and mental well-being. In addition to merely punishing for violent actions, it is crucial to deter and prevent their occurrence proactively. Implementing surveillance cameras in public spaces has effectively facilitated the prompt detection of emerging violent behavior. While this strategy has curbed violent incidents [1, 2], the widespread deployment of these cameras stirs up concerns over potential invasions of individuals' privacy, leading to significant apprehensions.

The proliferation of cameras has dramatically enhanced the ease of data collection. Cameras are commonly employed for indoor and outdoor surveillance, capturing instances of violence or emergencies [3, 4, 5]. Nonetheless, this data-gathering method frequently requires obtaining explicit consent from recorded participants for public data collection. Obtaining comprehensive consent from individuals captured on camera poses significant challenges [3, 6]. In addition to capturing movement data, personal information related to privacy, such as facial features and attire, is recorded and potentially stored on untrusted third-party servers with high-performance capabilities, thereby intensifying the potential for privacy breaches. Privacy refers to personal information that an individual does not wish

---

*These authors contributed equally.
†Corresponding author: yi.zeng@ia.ac.cn
 Project website: https://www.brain-cog.network/dataset/Bullying10k/
 Dataset website: https://figshare.com/articles/dataset/Bullying10k/19160663

37th Conference on Neural Information Processing Systems (NeurIPS 2023) Track on Datasets and Benchmarks.

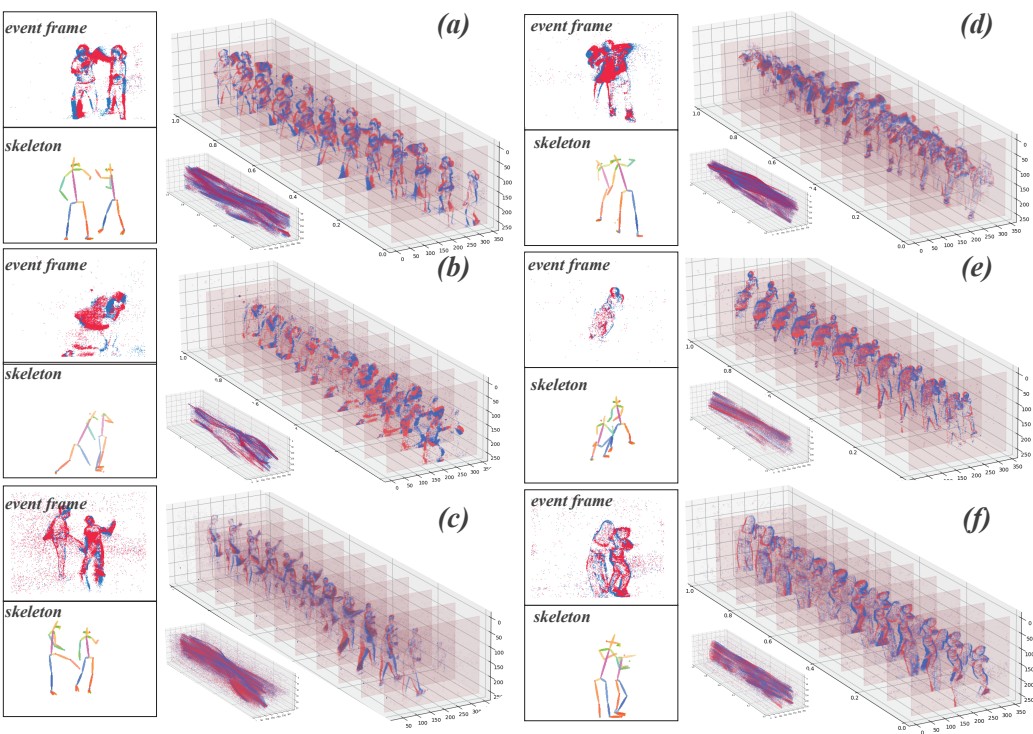

Figure 1: Visualization of the Bullying10K dataset. For each example, the right section illustrates the stream of events captured by a Dynamic Visual Sensor (DVS) camera, showcasing the dynamic changes in brightness at each pixel. The left section demonstrates the related event frame transformed from the event stream and corresponding human pose keypoints labels. This enables us to observe specific actions and scenes.

to be accessed or understood by others. The disclosure of such information might result in a loss of benefits or harm to the individual's interests. Privacy preserving involves measures to ensure that private information contained within data is either eliminated or made extremely difficult to extract. Facial data is one of the most commonly used types of information for identity recognition. It has been widely adopted for applications such as facial payment systems [7] and device unlocking [8], making it one of the most concerned about in terms of privacy. Therefore, in this article, the primary focus of privacy protection is on facial information used in critical payment and unlocking scenarios rather than broadly on whether an individual can be identified. At present, most of the commonly employed violence detection datasets primarily utilize RGB images. We aim to devise a strategy that effectively identifies unusual and violent incidents while minimizing the risk of privacy breaches during normal circumstances.

Dynamic Vision Sensors (DVS) cameras [9], which capture pixel brightness variations, provide an innovative alternative to conventional cameras that produce image frames at fixed frequencies. Instead, DVS cameras generate an event stream that records each pixel's brightness changes, either enhancement or reduction. As a result, it becomes challenging to visually identify the captured objects. Although some techniques attempt to reconstruct images from DVS data [10, 11], these approaches grapple with issues such as low contrast and blurriness and may even require additional sensor information for assistance [11]. Consequently, extracting detailed user information suitable for recognition systems from DVS cameras becomes a significant challenge, naturally reinforcing privacy persevering measures. At the same time, the high sensitivity of DVS cameras ensures their stable performance under uncontrolled luminary conditions and diverse environmental states [12]. As an event-driven camera, DVS consumes low power when the scene is static, reducing energy consumption and mitigating information redundancy compared to traditional cameras. However, although datasets captured using DVS cameras exist [13, 14, 15], most are employed for traditional image classification tasks. Existing action recognition datasets [16, 17] primarily focus on generic

Table 1: Comparison of different neuromorphic datasets.

| Dataset | #Year | #Sensor | #Type | #Object | #Sec Per Example | #pose | #Event Count |
|---|---|---|---|---|---|---|---|
| **ASLAN-DVS** [18] | 2019 | Davis240c | reproduced | action | - | No | - |
| **N-MNIST** [14] | 2015 | ATIS | reproduced | digit images | 0.3s | - | 300M/4K |
| **N-CALTECH101** [14] | 2015 | ATIS | reproduced | images | 0.3s | - | 1B/0.1M |
| **DVS-CIFAR10** [13] | 2017 | Davis128 | reproduced | images | 1.2s | - | 2B/0.2M |
| **HMDB-DVS** [18, 19] | 2019 | Davis240c | reproduced | action | 19s | No | 3B/0.5M |
| **ES-ImageNet** [20] | 2021 | - | conversion | images | - | - | - |
| **UCF-DVS** [18, 21] | 2019 | Davis240c | reproduced | action | 25s | No | 12B/0.9M |
| **N-Omniglot** [15] | 2022 | Davis346 | reproduced | char images | - | - | - |
| **DVS-Gesture** [17] | 2017 | Davis128 | real | action | 6s | No | 500M/0.4M |
| **NCARS** [22] | 2018 | ATIS | real | cars | 0.1s | - | 95M/4K |
| **ASL-DVS** [23] | 2020 | Davis240 | real | hand | 0.1s | - | 2B/21K |
| **PAF** [16] | 2019 | Davis346 | real | action | 5s | No | - |
| **DailyAction** [24] | 2021 | Davis346 | real | action | 5s | No | - |
| **Bullying10K** | 2023 | Davis346 | real | action | 2-20s | Yes | 12B/1.2M |

simple action recognition with limited scale and simplistic labels. Thus, they are insufficient for detecting complex and rapid actions and overlapping individuals, characteristic of violent incidents.

To address these concerns, we leverage the unique characteristics of DVS cameras and propose an event-based dataset called **Bullying10K**. The dataset aims to detect violent incidents in videos while ensuring privacy protection. Instead of relying on conversion algorithms or reproduction methods that can be time-saving and resource-saving, we chose to capture real-life scenarios and subjects using DVS cameras. This approach allows us to avoid data biases that may arise from the process of RGB cameras of original datasets. The dataset captures subjects engaging in various actions under different views and lighting conditions. In addition to data privacy persevering, the Bullying10K dataset stands out from other DVS datasets by encompassing more complex and rapid actions and instances where individuals may obscure each other. This inclusion introduces new challenges to event-based neuromorphic datasets.

In conclusion, the design of the Bullying10K dataset aims to fulfill the real-time detection requirements of violent behavior while maximizing the privacy protection of the individuals captured in the footage. This dataset will serve as valuable training data for developing privacy-preserving video systems, providing new insights and opportunities for future research.

Our contributions are as follows:

1. We propose a large-scale DVS bullying recognition dataset: Bullying10K. It contains 10,000 event segments, totaling 12 billion events and 255 GB of data. The actions in the videos are characterized by their complexity, rapidity, and occlusion of individuals.

2. We provide three benchmarks for comparing the performance of different methods: an action recognition benchmark, an temporal action localization benchmark, and a pose estimation benchmark. For the pose estimation task, we provide the keypoints of human pose.

3. We present the DVS community with a trainable dataset to detect violent scenes without compromising privacy. It makes the anticipation and research of violent scenarios possible.

## 2   Related Work

**DVS Dataset**    Early DVS datasets were typically derived from pre-existing image classification datasets [13, 14]. They captured the brightness differences of pixels caused by camera or image motion using DVS cameras. However, generating meaningful temporal data from static images proved challenging. [17, 16] captured people in real scenarios, showcasing different actions through hand movements and providing early event-based classification task benchmarks. However, these datasets were relatively small, and the actions displayed were somewhat repetitive. In contrast to traditional classification tasks, [15] introduced a dataset for few-shot tasks, reconstructing the drawing process of character strokes and creating meaningful temporal data. [18] attempted to capture existing action recognition datasets using DVS cameras. However, video reproduction failed to capture the event

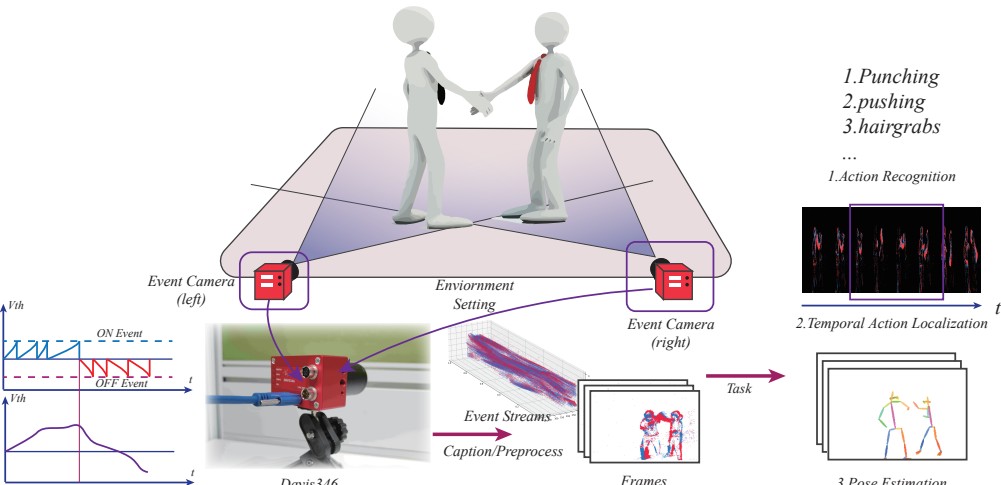

Figure 2: The flow of the data acquisition process. We employed two DVS cameras, positioned on the left and right sides, respectively. Following the recording, the DVS outputs an event stream for pre-processing. This processed data was then employed for three distinct tasks: action recognition, temporal action localization, and pose estimation.

characteristics in natural scenes, especially motion blur caused by high-speed motion or significant changes in illumination conditions. Constructing a dataset suitable for violence detection requires capturing data with complex actions, fast movement, and occlusion, which existing datasets are not explicitly designed for. In Table 1, we provide an overview of several DVS datasets, where #Event Count means the total number of events of the dataset and the average number of each example.

**Violent Dataset**  The construction of appropriate datasets for detecting violent actions are crucial to promote research in automated detection technology. [25] proposed a dataset created by extracting clips from short films comprising only 200 video segments. [26] collected 1,000 data samples by capturing snippets from hockey games. [27] gathered realistic scenes involving multiple groups engaged in specific actions. [28] compiled the RWF-2000 dataset by amassing 2,000 sample clips from the internet. [29] extracted segments from Hollywood movies to create a dataset.

**Spiking Neural Networks (SNNs)** are models that simulate the behavior of neurons in the brain. In contrast to Artificial Neural Networks (ANNs), SNNs transmit signals through discrete spikes, and the accumulation of membrane potentials in SNNs allows them to handle time series data effectively, making them well-suited for processing event-based data. However, due to the non-differentiability of spike sequences, applying the traditional backpropagation (BP) algorithm directly to training SNNs poses significant challenges. As a result, various methods have been proposed to explore effective training approaches for SNNs [30, 31, 32, 33, 34, 35, 36, 37, 38].

**Privacy-Preserving Action Recognition** Image or video editing stands as the most prevalent method of ensuring visual privacy. These methods can be broadly categorized into the following three: Filtering: This involves techniques like spatial downsampling, blurring, and pixelation of images [39, 40, 41, 42]. Intuitively, these methods are effective in preserving privacy. However, by treating all information uniformly, the removal of private data can significantly hinder accurate action recognition. Empirical Obfuscation: This method masks privacy-sensitive information irrelevant to the primary task. Examples include using object detection or segmentation to edit or eliminate faces or bodies [43, 44]. Its efficacy depends on the detector's performance and the adaptability between the target and source domains. Information outside of empirical information might still be left exposed. Learning-based Obfuscation: This approach balances task performance and privacy protection. It actively suppresses sensitive attributes within visual data via adversarial learning [45, 46, 47]. However, for effective privacy assurance, training such a model demands a substantial amount of computational power.

# 3 Bullying10K Dataset

In this section, we elaborate on the acquisition process, preprocessing methods, and annotation details of the Bullying10K dataset. Simultaneously, we analyze multiple attributes of the dataset, including its temporal length, keypoints motion, and spatial event distribution.

## 3.1 Data Acquisition

**Environment Setting** For data collection, we utilize two Davis346 [48], a high-speed event camera that captures pixel brightness changes with microsecond precision. Each pixel's brightness change triggers an event $(t, x, y, p)$, where $(x, y)$ represents the spatial coordinates of the pixel, $t$ denotes the event's time, and $p$ is either 0 or 1, indicating the polarity of the brightness change (enhancement or reduction). To capture multiple viewing angles and ensure diversity in the collected data, we position two DVS cameras on the left and right sides of the filming scene, as depicted in Figure 2. For consistency in the dataset, the cameras were positioned 5 meters apart, with both camera lenses were oriented at a 30-degree angle from the direct front. Additionally, To capture a diverse range of data and to more closely align with real-world conditions, we set up two lighting conditions: light and dark.

The sensitivity of the DVS cameras allows for a time precision of less than 1us, which results in more noise under dark conditions compared to light ones. The camera lenses have a focal length of 4mm, an aperture of 1.6, and an exposure time of 20ms to ensure an appropriate filming range and exposure. we invited 25 distinct participants, leading to 50 recording groups in total. The gender ratio among participants was 1:1. Instead of merely repeating a specific action, and participants were encouraged to execute movements freely while ensuring the action type remained consistent, adding to the dataset's diversity and complexity. Each event segment contains two participants who assume the roles of a perpetrator and a victim during segments involving violent actions. In contrast, sections depicting friendly

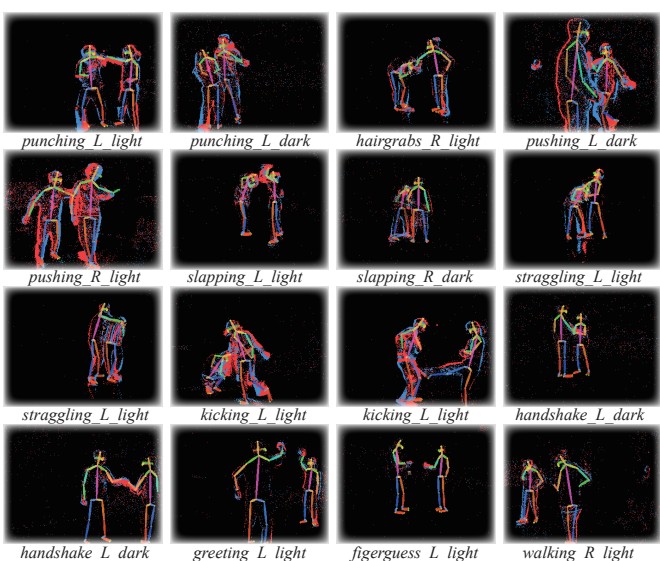

Figure 3: Visualization of human pose keypoints labels on event frames. The labels denote the action name, camera position, and lighting conditions.

actions involve the cooperation between the two participants. Actors are instructed to perform a specific action in each video segment, and we collect ten valid clips for each action. The duration of each sample segment is action-dependent, ranging from 2 to 20 seconds.

**Preprocessing** The Davis346 camera directly outputs data in the `aedat4` file format, specifically designed for storing event streams. To facilitate subsequent processing and analysis, we transform the raw data into the widely used `npy` format. `npy` is a common file format for storing NumPy [49] array data, enabling effortless preservation and recovery of multidimensional data, matrices, and other data structures. Throughout this transformation, we organized the event stream into 10-millisecond units to maintain temporal precision while effectively compressing the data. This strategy improves convenience and reduces file size. For user-friendly data manipulation, we supply code that merges the event stream into frames and reads the data.

**Quality Control** To ensure data quality and usability of the data, we marked the position of each camera before the start of filming, along with the relevant settings of the DVS camera (including aperture, focal length, etc.), and maintained consistency in these settings for each capture. To enhance

data redundancy and robustness, we introduced a manual screening step to optimize the data collection process. We captured twelve sample segments for each action group. Following data collection, we conducted manual screening to exclude poorly captured segments, using only ten segments per group for the final dataset. This ensures that every sample segment in the dataset possesses a good quality, facilitating subsequent research and analysis.

## 3.2 Data Annotation

### 3.2.1 Category Label

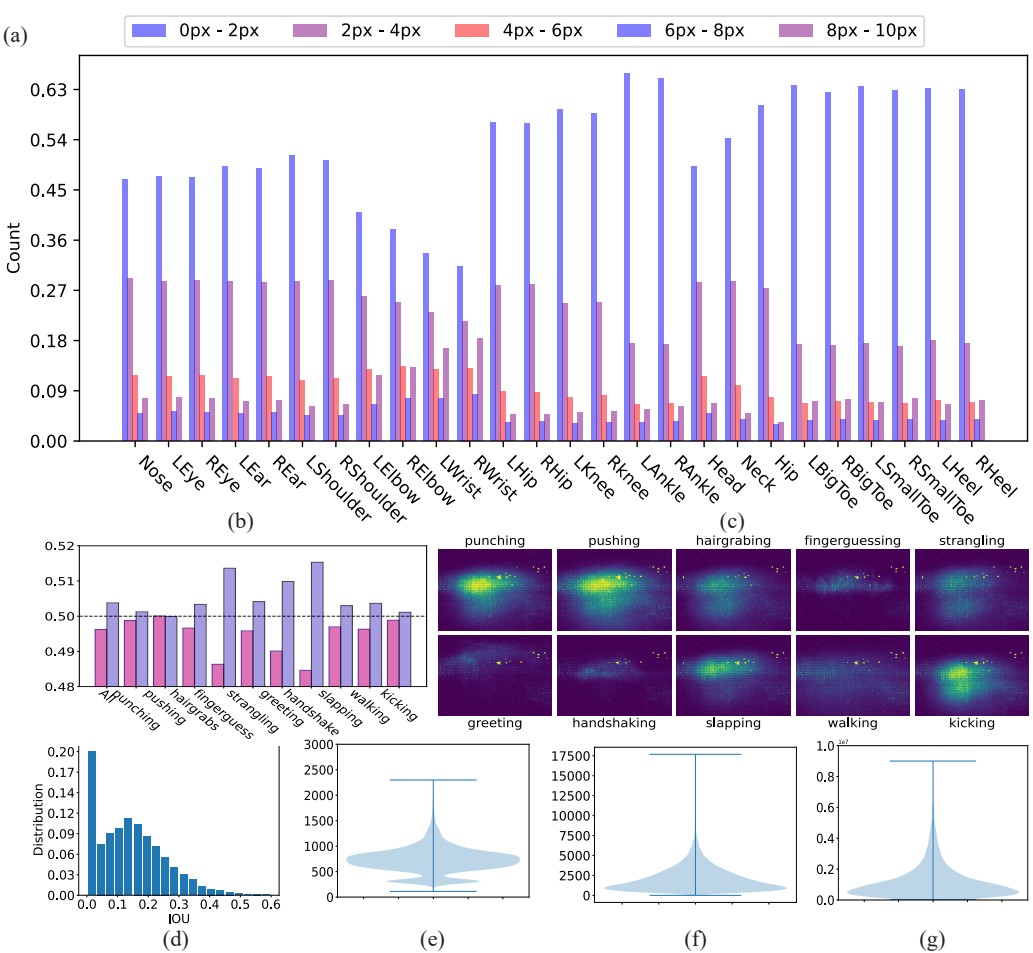

Figure 4: Statistical data and analysis of Bullying10K. (a) motion of keypoints. (b) the ratio of events in different polar. (c) distribution of events. (d) IoU of individual. (e) distribution of frame number. (f) distribution of event numbers. (g) event number in the sample.

We conduct detailed classification and annotation for each sample after filming for action recognition tasks to identify the represented action accurately. Our dataset consists of ten actions, including six violent actions (punching, kicking, hair grabbing, strangling, pushing, and slapping) and four friendly actions (handshaking, finger guessing, greeting, and walking). We further organized each category based on subjects, lighting scenes, and camera positions. Each group is named using the subject's code, action name, illumination, and camera position.

### 3.2.2 Pose Estimation

Pose estimation is a task that involves identifying a person's body position and keypoints in a video or an image. Precise pose estimation facilitates effective action recognition, making it an essential precursor to subsequent action recognition tasks. To acquire human pose data for each DVS video

segment, we simultaneously leverage the RGB data captured with the event data. The same camera captures both data types and offers overlapping scenes with highly consistent content. This advantage allows us to employ well-established pose estimation algorithms to predict the RGB dataset and obtain initial human pose labels. Specifically, we utilize AlphaPose [50] as an automated labeling tool, a multi-person pose estimation system. We employ the ResNet50 [51] backbone pre-trained on the Haple dataset, while the annotation process involved using the YOLOX [52] algorithm trained on the COCO dataset [53] as an object detector. Our annotation target includes 26 keypoints of the human body, as specified in [50], and the label information is saved in the COCO format. Upon obtaining initial labels, we manually calibrated the labels, as illustrated in Figure 3. It is important to note that direct pose estimation algorithms for DVS data are still in the early stages of development. Moreover, due to the inherent characteristics of DVS data, the events do not explicitly represent human poses, which compounds the complexity of directly utilizing DVS data for pose annotation.

## 3.3 Data Analysis

The Bullying10K dataset encompasses 10,000 sample clips, each with a duration ranging from 2 to 20 seconds. It contains an amount of 12 billion valid events, resulting in a total data volume of 255 GB. Figure 4 (e,f,g) presents the distribution of frames, events, and events per frame in each sample clip within the dataset. Notably, the average sample length of Bullying10K surpasses existing event datasets captured with DVS cameras. This addresses the limitation of shorter datasets captured by DVS cameras, providing new challenges in establishing long-range dependencies between event data.

Figure 4 (a) displays the results obtained by analyzing the movement distance of keypoints between consecutive frames. Different keypoints exhibit distinct motion distribution patterns, with the most prominent trends observed in the wrist and elbow movements, aligning with human motion characteristics. To quantify the degree of task overlap in the video, we calculate the Intersection over Union (IoU) of the bounding boxes for two characters appearing in the same frame. Figure 4 (d) visualizes the distribution of IoU values, which are primarily concentrated between 0.1 and 0.5. This indicates a significant number of instances where characters overlap within this dataset.

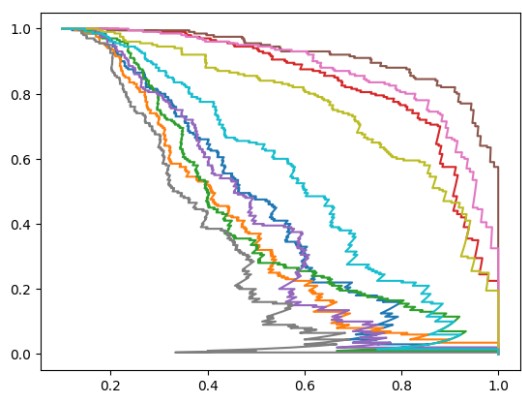

Figure 5: PR curves for each category obtained from the X3D model trained on Bullying10k.

As illustrated in Figure 4 (b), our analysis reveals a slightly lower occurrence ratio of positive and negative polarity events. Moreover, we visualize the ratio of positive and negative events for each action category within the dataset. As shown in Figure 4 (c). The movement distribution of actions such as punching and strangling predominantly occurs in the upper part of the image, while walking exhibits a higher likelihood of occurring in both the upper and lower parts of the image.

## 4 Evaluation and Task

We have provided three benchmark tasks: action recognition, temporal action localization, and pose estimation.

### 4.1 Action Recognition

**Experimental Setting**  The action recognition task aims to predict the corresponding action labels based on input images. Each sample segment in the dataset includes a single behavior, making it suitable for single-label classification tasks. During preprocessing, the event streams are combined into frames, forming a sequence that is then used to perform the action. The Bullying10K dataset has been divided into training and validation sets with an 8:2 ratio, and the temporal unit for integrating the event stream is set to 10 ms. We provided a corresponding interface in the code to ensure data consistency. Given many frames and the need for batch training, we randomly crop video

Table 2: Performance of different action recognition models on Bullying10k.

| Model | backbone | step 4 | | | step 10 | | | step 16 | | |
|---|---|---|---|---|---|---|---|---|---|---|
| | | gap 0 | gap 2 | gap 4 | gap 0 | gap 2 | gap 4 | gap 0 | gap 2 | gap 4 |
| C3D [55] | Conv3D | 47.60 | 49.05 | 52.10 | 51.70 | 54.75 | 57.20 | 60.35 | 68.55 | 71.25 |
| TAM [56] | ResNet | 54.85 | 59.20 | 62.00 | 57.85 | 60.80 | 65.05 | 58.20 | 66.90 | 71.20 |
| R2Plus1D [57] | ResNet18 | 54.60 | 57.15 | 60.15 | 56.50 | 61.60 | 65.45 | 58.20 | 65.70 | 69.25 |
| R3D [58] | ResNet18 | 57.20 | 61.05 | 62.10 | 58.90 | 65.46 | 68.60 | 62.70 | 69.75 | 72.50 |
| SlowFast [59] | ResNet50 | 57.80 | 60.30 | 61.55 | 60.10 | 66.50 | 70.90 | 61.70 | 70.55 | 74.00 |
| X3D [60] | ResNet | 60.30 | 62.40 | 64.80 | 63.30 | 69.40 | 72.15 | 65.75 | 72.45 | 76.90 |
| SNN [61] | SEW-ResNet19 | 51.75 | 53.00 | 53.85 | 56.40 | 59.50 | 62.85 | 58.48 | 64.05 | 67.05 |

Table 3: Performance of different privacy-preserving action recognition models on Bullying10k.

| Model | RGB | DVS | DS-2 [47] | DS-4 [47] | GB-3 [45] | GB-5 [45] | BDQ-1 [62] | BDQ-2 [62] |
|---|---|---|---|---|---|---|---|---|
| R3D [58] | 64.00 | 66.80 | 63.30 | 63.15 | 62.70 | 61.45 | 60.10 | 59.75 |
| SlowFast [59] | 59.25 | 69.00 | 57.80 | 55.40 | 57.20 | 54.95 | 60.20 | 59.45 |
| X3D [60] | 63.20 | 70.80 | 60.75 | 52.25 | 58.20 | 47.80 | 67.15 | 65.60 |

segments to a fixed length, sampling at 0, 2, and 4 intervals. We employed various commonly used action recognition models to evaluate the performance of the Bullying10K dataset. We explored the performance of spiking neural networks on this dataset, showcasing their potential for action recognition tasks.

**Evaluation Metric**    In the classification task, we measure the performance of the network by calculating the accuracy of the output corresponding to the labels. However, misjudgment of violent events can have severe consequences and cause irreparable harm. Violent scenarios occur less frequently than non-violent ones, implying a substantial class imbalance in real-world situations. Relying solely on accuracy as an evaluation metric may not adequately reflect the model's ability to predict violent events accurately. To address this issue, we have employed the Precision-Recall (PR) curve [54]. The PR curve allows us to examine the model's predictive capabilities across varying judgment thresholds.

**Results and Analysis**    Table 2 presents a detailed comparison of the performance of several widely used action recognition models on the Bullying10K dataset. It includes the backbone architectures of each model and their respective operational configurations. We conducted experiments using frame intervals of 0, 2, and 4, with three different temporal step lengths of 4, 10, and 16. The results demonstrate that increasing the frame interval and temporal step length contributes to improved precision of the models. However, even models that exhibit strong performance in traditional visual classification tasks did not yield satisfactory results on the Bullying10K dataset, indicating the dataset's unique challenges. Furthermore, performed a visual analysis of the corresponding PR curve of the X3D model on the Bullying10K dataset. The PR curve reveals significant variations in performance across different action categories, underscoring the dataset's complexity and the need for robust recognition models.

We implemented different privacy protection methods on the corresponding RGB frames of the Bullying10k dataset. We observed that after employing privacy protection with RGB data, there is a decline in performance. However, DVS data demonstrated slightly superior results compared to RGB data. DVS inherently offers advantages such as resilience to significant illumination changes, motion blur resistance, and reduced data redundancy. Therefore, for tasks like action recognition that rely on motion characteristics, the features captured by DVS are particularly beneficial.

### 4.2    Temporal action localization

**Experimental Setting**    In surveillance videos, violent scenarios often occur irregularly and sporadically. Due to the few background disturbances and noise in event data, we can naturally concatenate multiple samples to form extended sequences. In our task, we input a video frame sequence that encompasses segments from different action categories randomly extracted from the dataset and

Table 4: Performance of different temporal action localization models on Bullying10k.

| Model | Feature | $AUC$ | $AR@1$ | $AR@5$ | $AR@10$ | $AR@100$ |
|---|---|---|---|---|---|---|
| BSN [67] | TSN [63] | 75.9 | 31.1 | 70.5 | 75.8 | 77.6 |
| BSN | TSM [64] | 75.5 | 31.8 | 71.2 | 75.8 | 77.2 |
| BMN [68] | TSN | 80.3 | 34.6 | 74.3 | 81.0 | 82.0 |
| BMN | TSM | 81.0 | 36.0 | 75.3 | 82.4 | 82.7 |

Table 5: Performance of different pose recognition models on Bullying10k.

| Model | SimpleBaseline [69] (ResNet-50) | SimpleBaseline [69] (ResNet-101) | HRNet [70] (HRNet-ws32) | HRNet [70] (HRNet-ws48) | SimpleBaseline [69] (Spiking ResNet-50) |
|---|---|---|---|---|---|
| AP | 62.6 | 63.6 | 62.7 | 62.8 | 54.1 |
| $AP^{50}$ | 88.3 | 88.2 | 88.2 | 87.8 | 84.6 |
| $AP^{75}$ | 67.4 | 67.7 | 67.5 | 67.2 | 57.6 |
| $AP^{M}$ | 58.3 | 59.3 | 58.6 | 59.5 | 49.3 |
| $AP^{L}$ | 73.4 | 74.6 | 73.9 | 74.9 | 64.9 |
| AR | 65.9 | 67.0 | 66.3 | 66.7 | 58.5 |

integrate them into longer video sequences. And aim to predict the action label along with its corresponding start and end times. The ground truth annotations are constructed based on the respective time intervals. For training on the Bullying10K dataset, we have chosen commonly used temporal action localization models. To achieve more accurate testing results, we provided action recognition data for the pre-trained features at the same ratio. Additionally, we trained the model for the localization task at a 1:3 ratio. Initially, we pre-trained the TSN [63] and TSM [64] models on the action recognition task, which serve as the feature extractors for the localization models. We extract features from the dataset and store them for subsequent analysis and processing.

**Evaluation Metric**    Average recall [65] and Area Under the Curve (AUC) [66] are commonly used metrics to evaluate action localization. Concurrently, we employ the AR@N indicator for assessment, representing the recall rate under the condition of N proposals. This study considers N to be 1, 5, 10, and 100. Additionally, we calculate the AUC for the AR-AN curve.

**Results and Analysis**    We present the accuracy of several commonly used temporal action localization algorithms on Bullying10K for comparative evaluation. They exhibit varying performances under different features. On the other hand, performance significantly diminishes with the reduction of proposals. It is worth exploring a model designed for processing event datasets that can maintain higher precision with fewer proposals.

### 4.3   Pose estimation

**Experimental Setting**    Human pose estimation typically involves providing an image or video containing one or multiple individuals and outputting the corresponding locations of various keypoints for each person. These keypoints include the head, shoulders, arms, and legs. Different datasets may have different numbers of joints, and in the case of Bullying10K, we use 26

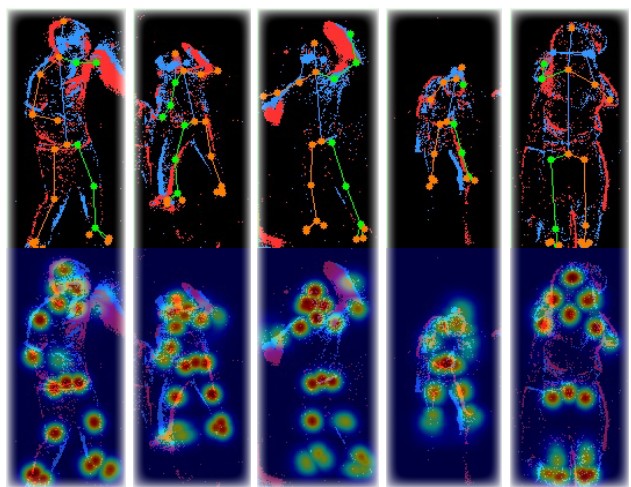

Figure 6: Taking the prediction results of the SimpleBaseline model with ResNet101 as the backbone, we visualize the skeleton and heatmap of the prediction results, respectively.

keypoints. Pose estimation can provide additional information for action recognition. We split the dataset at an 8:2 ratio and have provided the corresponding pose annotations along with the data to ensure consistency. We evaluate the performance of Bullying10K using several commonly used pose estimation models.

**Evaluation Metric**   The standard metric used to measure pose estimation is usually based on Object Keypoint Similarity (OKS) [71]. OKS is a scale-invariant measure of localization accuracy and is utilized to evaluate how closely the predicted keypoints of a model match the ground truth keypoints. $OKS = \frac{\sum_i [\exp(-d_i^2/2s^2 k_i^2)\delta(v_i > 0)]}{\sum_i [\delta(v_i > 0)]}$, where $d_i$ is the distance between the position of predicted keypoints and the ground truth. $v_i$ denotes the visibility of related keypoints. $s$ indicates the object scale. $k_i$ is the predefined constant that controls falloff. Average Precision (AP) and Recall (AR) are used, including $AP$, $AP^{50}$ (OKS is at 0.50), $AP^{75}$, $AP^M$, $AP^L$, $AR$ to measure the accuracy of different models on the Bullying10K dataset.

**Results and Analysis**   Table 5 presents the results of multiple commonly used pose estimation models on the Bullying10K dataset. However, these models exhibit relatively low accuracy on the dataset, indicating that Bullying10K poses substantial challenges for pose estimation tasks. Additionally, although HRNet has shown superior capabilities to SimpleBaseline on other RGB datasets, it achieved lower accuracy on our dataset. This discrepancy may be attributed to the unique nature of our event-based dataset, which significantly differs from RGB images. Furthermore, we investigated the potential of SNN in posture estimation by testing the SNN backbone using the SimpleBaseline algorithm. To enhance the interpretability of the results, we visualized relevant output images and presented heatmaps for selected samples.

## 5   Discussion

This research introduces a novel event-driven dataset called Bullying10K, which utilizes Dynamic Vision Sensor (DVS) cameras to detect instances of violent behavior while preserving individual privacy. This approach offers a novel possibility for privacy preservation, distinct from traditional image surveillance methods. It has an influence in the field of privacy protection and security surveillance.

However, typically, the kind of private information that cameras can capture is varied and encompasses aspects like facial details, gait patterns, and even individuals' habits of daily life. Some recent technologies can identify specific individuals using non-facial information [72, 73, 74]. Our dataset might struggle to prevent leaks of non-facial information, such as gait data. Still, we wish to emphasize that facial data is the most commonly used and has been incorporated into many crucial applications. This type of privacy is our primary focus in this paper.

Moreover, for the reason for clarity in labeling and for comparability during model validation, our dataset defines specific, commonly observed actions. It's challenging to cover every possible violent or non-violent action a person might exhibit comprehensively. However, an incomplete category set might increase the risk of classification errors in the model, which could adversely affect judgments regarding violence.

The dataset addressed the limitations of existing event-driven datasets by featuring complex, rapid movements and overlapping figures, presenting higher complexity and challenges. By offering a large-scale dataset, Bullying10K enables researchers to explore complex actions and contributes to advancements in violence detection and privacy-preservation techniques.

## Acknowledgments and Disclosure of Funding

This paper is funded by the National Key Research and Development Program (Grant No. 2020AAA0107800).

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
