# A  The Bullying10K Dataset Datasheet

## A.1  Motivation

**1. For what purpose was the dataset created? Was there a specific task in mind? Was there a specific gap that needed to be filled? Please provide a description.**

A1: The Bullying10K dataset is a neuromorphic dataset used for violence detection. As surveillance cameras become prevalent in public spaces, using them has proven effective in proactively deterring and preventing such incidents. However, the data collected by these cameras could potentially lead to breaches in privacy for those being filmed. Thus, we hope to find a way to capture scenes of violence while avoiding infringement on personal privacy. DVS cameras can naturally achieve this goal by capturing events of pixel brightness changes. Existing violence detection datasets are filmed with RGB cameras, which cannot ensure privacy preserving. Furthermore, existing DVS datasets do not contain complex and rapid actions that can be used for violence detection tasks. Therefore, we propose the Bullying10K to fulfill this task. Bullying10K provides trainable data for violence detection tasks based on privacy preserving. We also provide keypoints labels for pose estimation for Bullying10K, which can aid in understanding human action recognition under neuromorphic data.

**2. Who created this dataset (e.g., which team, research group) and on behalf of which entity (e.g., company, institution, organization)?**

A2: Bullying10K is created by Yiting Dong, Yang Li, Dongcheng Zhao, Guobin Shen, and Yi Zeng from the Brain-inspired Cognitive Intelligence Lab, Institute of Automation, Chinese Academy of Sciences.

**3. Who funded the creation of the dataset? If there is an associated grant, please provide the name of the grantor and the grant name and number.**

A3: This paper is funded by the National Key Research and Development Program (Grant No. 2020AAA0107800).

## A.2  Composition

**1. What do the instances that comprise the dataset represent (e.g., documents, photos, people, countries)? Are there multiple types of instances (e.g., movies, users, and ratings; people and interactions between them; nodes and edges)? Please provide a description.**

A1: Each instance is an event stream, which includes brightness change events for each pixel during the filming process.

**2. How many instances are there in total (of each type, if appropriate)?**

A2: The Bullying10K dataset has 10000 event stream with 12 billion events. We also provide related category labels and keypoints annotation for pose estimation.

**3. Does the dataset contain all possible instances or is it a sample (not necessarily random) of instances from a larger set? If the dataset is a sample, then what is the larger set? Is the sample representative of the larger set (e.g., geographic coverage)? If so, please describe how this represent ativeness was validated/verified. If it is not representative of the larger set, please describe why not (e.g., to cover a more diverse range of instances because instances were withheld or unavailable).**

A3: The Bullying10K is completely captured from scratch, and it is not sampled from a exist bigger dataset.

**4. What data does each instance consist of? "Raw" data (e.g., unprocessed text or images)or features? In either case, please provide a description.**

A4: Each instance is an event stream which contains multiple events. Each event is composed of $(x, y, p, t)$, which represents the brightness changes of the pixel (enhanced and decreased). Where $x, y$ represent the position of the pixel, $p$ represents polarity, and $t$ represents the timestamp of the event. Each event stream is saved in the npy format.

**5. Is there a label or target associated with each instance? If so, please provide a description.**

A5: Yes, Bullying10K provides a category label for each instance, including action name, DVS camera position, illumination, and individuals' code. It also provides keypoints for pose estimation, which has 26 keypoints for each individual.

**6. Is any information missing from individual instances? If so, please provide a description, explaining why this information is missing (e.g., because it was unavailable). This does not include intentionally removed information, but might include, e.g., redacted text.**

A6: Yes, during our annotation process, given that the frames are composed of events, they lack the actual location of human keypoints. Also, due to the occlusion of individuals, we are unable to label the overlapped positions. We have set all unlabelable positions as unavailable.

**7. Are relationships between individual instances made explicit (e.g., users' movie ratings, social network links)? If so, please describe how these relationships are made explicit.**

A7: Yes, we have adopted detailed category annotations and the COCO annotation format for pose estimation. This allows us to understand the relationships between different event streams, such as the correspondence of event streams to left and right positions.

**8. Are there recommended data splits (e.g., training, development/validation, testing)? If so, please provide a description of these splits, explaining the rationale behind them.**

A8: Yes, we split the dataset with a ratio of 0.8 to 0.2. To ensure that there's no significant bias in the distribution between the training and testing sets, we sample the same number of samples from each category.

**9. Are there any errors, sources of noise, or redundancies in the dataset? If so, please provide a description.**

A9: Even with the introduction of manual inspection, due to the nature of the events and issues with occlusion, some keypoints might not be able to be annotated.

**10. Is the dataset self-contained, or does it link to or otherwise rely on external resources (e.g., websites, tweets, other datasets)? If it links to or relies on external resources, a) are there guarantees that they will exist and remain constant over time; b) are there official archival versions of the complete dataset (i.e., including the external resources as they existed at the time the dataset was created); c) are there any restrictions (e.g., licenses, fees) associated with any of the external resources that might apply to a future user? Please provide descriptions of all external resources and any restrictions associated with them, as well as links or other access points, as appropriate.**

A10: Yes, the Bullying10K is completely captured from scratch, and it doesn't rely on any external resources.

**11. Does the dataset contain data that might be considered confidential (e.g., data that is protected by legal privilege or by doctorpatient confidentiality, data that includes the content of individuals non-public communications)? If so, please provide a description.**

A11: No, we have taken measures such as wearing masks to minimize the risk of privacy breaches.

**12. Does the dataset contain data that, if viewed directly, might be offensive, insulting, threatening, or might otherwise cause anxiety? If so, please describe why.**

A12: No, although the subject of our data is violence detection, it can not directly observe the content because it uses the DVS camera and the result was a stream of events. The content can be observed indirectly by combining the events into frames.

**13. Does the dataset relate to people? If not, you may skip the remaining questions in this section.**

A13: Yes

**14. Does the dataset identify any subpopulations (e.g., by age, gender)? If so, please describe how these subpopulations are identified and provide a description of their respective distributions within the dataset.**

A14: No

**15. Is it possible to identify individuals (i.e., one or more natural persons), either directly or indirectly (i.e., in combination with other data) from the dataset? If so, please describe how.**

A15: There is a possibility of identifying individuals through information such as their gait.

The kind of private information that cameras can capture is varied and encompasses aspects like facial details, gait patterns, and even individuals' habits of daily life. Some recent technologies can identify specific individuals using non-facial information. Our dataset might struggle to prevent leaks of non-facial information, such as gait data. [1, 2, 3]

**16. Does the dataset contain data that might be considered sensitive in any way (e.g., data that reveals racial or ethnic origins, sexual orientations, religious beliefs, political opinions or union memberships, or locations; financial or health data; biometric or genetic data; forms of government identification, such as social security numbers; criminal history)? If so, please provide a description.**

A16: No

## A.3    Collection Process

**1. How was the data associated with each instance acquired? Was the data directly observable (e.g., raw text, movie ratings), reported by subjects (e.g., survey responses), or indirectly inferred/derived from other data (e.g., part-of-speech tags, model-based guesses for age or language)? If data was reported by subjects or indirectly inferred/derived from other data, was the data validated/verified? If so, please describe how.**

A1: Each data instance is a stream of events, and the subject cannot be directly observed from the stream of event. By combining events into frames, the subject can be indirectly observed.

**2. What mechanisms or procedures were used to collect the data (e.g., hardware apparatus or sensor, manual human curation, software program, software API)? How were these mechanisms or procedures validated?**

A2: We use DVS cameras for capturing, which output events based on changes in pixel intensity. The camera is equipped with corresponding software that allows us to operate and capture. The sensitivity of the DVS cameras allows for a time precision of less than 1us, which results in more noise under dark conditions compared to light ones. The camera lenses have a focal length of 4mm, an aperture of 1.6, and an exposure time of 20ms to ensure an appropriate filming range and exposure.

**3. If the dataset is a sample from a larger set, what was the sampling strategy (e.g., deterministic, probabilistic with specific sampling probabilities)?**

A3: The Bullying10K is completely captured from scratch,

**4. Who was involved in the data collection process (e.g., students, crowdworkers, contractors) and how were they compensated (e.g., how much were crowdworkers paid)**

A4: A group of student volunteers participated as subjects in the filming, and we paid them that exceeded the local minimum wage standard with 100 RMB per hour and a total of 10000 RMB for the whole dataset. we invited 25 distinct participants, leading to 50 recording groups in total. The gender ratio among participants was 1:1.

**5. Over what timeframe was the data collected? Does this timeframe match the creation timeframe of the data associated with the instances (e.g., recent crawl of old news articles)? If not, please describe the timeframe in which the data associated with the instances was created.**

A5: The shooting period lasted for 3 months, while the data collection period spanned 1 month.

**6. Were any ethical review processes conducted (e.g., by an institutional review board)? If so, please provide a description of these review processes, including the outcomes, as well as a link or other access point to any supporting documentation.**

A6: No, but these streams of events were filmed with the subject's consent.

**7. Does the dataset relate to people? If not, you may skip the remainder of the questions in this section.**

A7: Yes

**8. Did you collect the data from the individuals in question directly, or obtain it via third parties or other sources (e.g., websites)? Were the individuals in question notified about the data collection? If so, please describe (or show with screenshots or other information) how notice was provided, and provide a link or other access point to, or otherwise reproduce, the exact language of the notification itself.**

A8: Yes, The subjects volunteered to participate and gave detailed instructions before shooting.

**9. Did the individuals in question consent to the collection and use of their data? If so, please describe (or show with screenshots or other information) how consent was requested and provided, and provide a link or other access point to, or otherwise reproduce, the exact language to which the individuals consented.**

A9: Yes, the filming was done with the consent of the subjects. The subjects volunteered to participate and gave detailed instructions before shooting.

**10. If consent was obtained, were the consenting individuals provided with a mechanism to revoke their consent in the future or for certain uses? If so, please provide a description, as well as a link or other access point to the mechanism (if appropriate).**

A10: Although we have obtained the permission of all the individuals, we still allow anyone to withdraw their permission. We will easily agree to delete the corresponding data of the individual from the dataset.

**11. Has an analysis of the potential impact of the dataset and its use on data subjects (e.g., a data protection impact analysis) been conducted? If so, please provide a description of this analysis, including the outcomes, as well as a link or other access point to any supporting documentation.**

A11: No

## A.4 Preprocessing/cleaning/labeling

**1. Was any preprocessing/cleaning/labeling of the data done (e.g., discretization or bucketing, tokenization, part-of-speech tagging, SIFT feature extraction, removal of instances, processing of missing values)? If so, please provide a description. If not, you may skip the remainder of the questions in this section.**

A1: After capturing, we manully check the event streams and annotations.

**2. Was the "raw" data saved in addition to the preprocessed/cleaned/labeled data (e.g., to support unanticipated future uses)? If so, please provide a link or other access point to the "raw" data.**

A2: We provide the event data which is the same as raw data.

**3. Is the software used to preprocess/clean/label the instances available? If so, please provide a link or other access point.**

A3: No

## A.5 Uses

**1. Has the dataset been used for any tasks already? If so, please provide a description.**

A1: Yes, In this paper, we have used it in tasks, action recognition, temporal action localization, pose estimation.

**2. Is there a repository that links to any or all papers or systems that use the dataset? If so, please provide a link or other access point.**

A2: N/A

**3. What (other) tasks could the dataset be used for?**

A3: The Bullying10K dataset can be used for topics such as action behavior understanding or prediction.

**4. Is there anything about the composition of the dataset or the way it was collected and preprocessed/cleaned/labeled that might impact future uses? For example, is there anything that a future user might need to know to avoid uses that could result in unfair treatment of individuals or groups (e.g., stereotyping, quality of service issues) or other undesirable harms (e.g., financial harms, legal risks) If so, please provide a description. Is there anything a future user could do to mitigate these undesirable harms?**

A4: Yes.

Some recent technologies can identify specific individuals using non-facial information. Our dataset might struggle to prevent leaks of non-facial information, such as gait data.

Moreover, for the reason for clarity in labeling and for comparability during model validation, our dataset defines specific, commonly observed actions. It's challenging to cover every possible violent or non-violent action a person might exhibit comprehensively. However, an incomplete category set might increase the risk of classification errors in the model, which could adversely affect judgments regarding violence.

**5. Are there tasks for which the dataset should not be used? If so, please provide a description.**

A5: Yes. Users should not attempt to identify participants using other methods in the future. Additionally, caution should be exercised when using datasets for the task of recognizing violent actions to prevent over-reliance on the model, which could lead to misclassification.

## A.6 Distribution

**1. Will the dataset be distributed to third parties outside of the entity (e.g., company, institution, organization) on behalf of which the dataset was created? If so, please provide a description.**

A1: We have distributed the Bullying10K dataset.

**2. How will the dataset will be distributed (e.g., tarball on website, API, GitHub)? Does the dataset have a digital object identifier (DOI)?**

A2: It have been available on the website https://figshare.com/articles/dataset/Bullying10k/19160663 or https://doi.org/10.6084/m9.figshare.19160663.

**3. When will the dataset be distributed?**

A3: We have distributed the Bullying10K dataset.

**4. Will the dataset be distributed under a copyright or other intellectual property (IP) license, and/or under applicable terms of use (ToU)? If so, please describe this license and/or ToU, and provide a link or other access point to, or otherwise reproduce, any relevant licensing terms or ToU, as well as any fees associated with these restrictions.**

A4: Bullying10K will be distributed under the CC-BY-4.0 license.

**5. Have any third parties imposed IP-based or other restrictions on the data associated with the instances? If so, please describe these restrictions and provide a link or other access point to, or otherwise reproduce, any relevant licensing terms, as well as any fees associated with these restrictions.**

A5: No

**6. Do any export controls or other regulatory restrictions apply to the dataset or to individual instances? If so, please describe these restrictions, and provide a link or other access point to, or otherwise reproduce, any supporting documentation.**

A6: No

## A.7 Maintenance

**1. Who will be supporting/hosting/maintaining the dataset?**

A1: The Authors.

**2. How can the owner/curator/manager of the dataset be contacted (e.g., email address)?**

A2: Contact us by email or our website.

**3. Is there an erratum? If so, please provide a link or other access point.**

A3: No, We have checked the data manually.

**4. Will the dataset be updated (e.g., to correct labeling errors, add new instances, delete instances)? If so, please describe how often, by whom, and how updates will be communicated to users (e.g., mailing list, GitHub)?**

A4: No

**5. If the dataset relates to people, are there applicable limits on the retention of the data associated with the instances (e.g., were individuals in question told that their data would be retained for a fixed period of time and then deleted)? If so, please describe these limits and explain how they will be enforced.**

A5: No

**6. Will older versions of the dataset continue to be supported/hosted/maintained? If so, please describe how. If not, please describe how its obsolescence will be communicated to users.**

A6: N/A

**7. If others want to extend/augment/build on/contribute to the dataset, is there a mechanism for them to do so? If so, please provide a description. Will these contributions be validated/verified? If so, please describe how. If not, why not? Is there a process for communicating/distributing these contributions to other users? If so, please provide a description.**

A7: N/A

## B  Data Access

We have released our dataset and in order to ensure its long-term preservation, we have uploaded it to the Figshare website. Figshare is a dedicated public platform for storing data. It is available on the website https://figshare.com/articles/dataset/Bullying10k/19160663 or using DOI https://doi.org/10.6084/m9.figshare.19160663.

You can also visit our website for more detailed information.https://www.brain-cog.network/dataset/Bullying10k/ We also publish the code for reading and processing the data. You can ask for the code on github. https://github.com/Brain-Cog-Lab/Bullying10K

## C  Statement of Responsibility

The authors declare that they bear all responsibility for violations of rights and that this dataset is released under CC-BY-4.0 license.

During the filming process, we adhered to the Personal Information Protection Law of the People's Republic of China (China's PIPL) and explained all privacy-related details to the participants. We ensured the participants were fully informed and obtained the consent of all involved.

## D  Hosting and Maintenance Plan

The authors and Brain-inspired cognitive intelligence lab will host the dataset and handle maintenance concerns. We release the Bullying10K dataset on the Figshare. You can download it from the website.

## E  Additional Detail

In all of our experiments, we utilized two A100 graphics cards for computations with the Ubuntu operating system, and all of our code is based on the PyTorch framework. Because of the multiple

tasks and experiments, for specific training parameters, refer to the configuration files within the code or default settings.

**Instruction**    We provided instructions that were given to the subjects before the filming began.

We will now begin with the experiment. Initially, two cameras will be used for filming. Please ensure that you stay within the camera's field of view when you are being filmed. There are a total of 10 different actions: punching, hair grabbing, kicking, pushing, slapping, handshaking, greeting, finger guessing, and walking. Each action will be filmed 12 times, with one set of filming under bright conditions and another under dark conditions. Once the experimenter indicates the start, the cameras will begin recording and the subject will begin the assigned action. The recording will conclude when the experimenter indicates the end, at which point the subject can stop their action. The duration of each recording will vary. During each recording, only the assigned action should be performed; switching between different actions within the same recording is not permitted. Feel free to ask the experimenter if you have any questions during the filming process.

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

## F    Photography Informed Consent Form

We provide the **Photography Informed Consent Form**, and translate it into English.

# Photography Informed Consent Form

Dear Participant,

Thank you for choosing to participate in our project. This informed consent is designed to ensure you are fully aware of all relevant information, risks, and rights associated with participating in this project.

**Project Description:**

We are developing a dataset designed for privacy-preserving detection of violent actions. Our filming will involve participants performing specified violent and non-violent actions. The filming will be done using a DVS (Dynamic Vision Sensor) camera, a type of camera that captures pixel brightness changes, enabling it to record movement information of the participants.

**Shooting Procedure:**

The videographer will guide you through the actions. You will perform a series of specified actions alongside another participant in front of two DVS cameras. The cameras will capture both RGB image information and motion event data. This includes:

  Violent actions: Punching, kicking, hair grabbing, strangling, pushing, and slapping.

  Friendly actions: Handshaking, finger guessing, greeting, and walking.

Each action will be filmed 12 times, under both light and dark conditions. The start and end of each action based on the videographer's instructions. The entire shoot will last approximately 1 to 2 hours.

**Data Usage:**

Your image data will only be used for research purposes. In any published research or dataset, we will not disclose any personal information beyond motion event data. Only motion event information will be released, without the accompanying RGB data. If additional data needs to be released, we will seek your consent once again.

**Your Rights:**

Participation in this project is entirely voluntary. You have the right to withdraw your consent and stterminate op participating at any time, for any reason. Your data will then be removed from the dataset. As an appreciation for your participation, you will receive a 100 RMB/h compensation.

**Risk Acknowledgement:**

While we do not disclose your personal information and the DVS camera obscures facial recognition, there are still techniques, without facial information, that might potentially identify individuals. It is essential you are aware of these risks before participating. The shoot involves significant physical movement, and there's a risk of injury during actions. Please exercise caution.

By signing below:

You have fully understood all the information presented above.

You have opportunity to ask any questions about the project ,have received satisfactory answers.

You are participating in this project of your own free will.

You consent to the data's release and publicity.

You understand that you can withdraw your consent and leave the study at any time.

Signature: _______________________________

Date: _________________________________