# OpenReview forum: "Bullying10K: A Large-Scale Neuromorphic Dataset towards Privacy-Preserving Bullying Recognition"
_NeurIPS.cc/2023/Track/Datasets_and_Benchmarks — NeurIPS 2023 Datasets and Benchmarks Poster_

### Official Review · Reviewer_dVaU · 2023-06-28
**Shouldn't be a "NeurIPS-labelled reference dataset" due to too many uncertainties and medium-to-minor flaws**

**Rating:** 6
**Confidence:** 4
**Clarity:** The paper is mostly well-written (eve…

**Strengths:**

- Substantial dataset
- Technically, the paper is mostly reasonable, but not overwhelming.

**Additional Feedback:**

Some additional minor things:

* "privacy preserving" is not a noun. Either say "privacy" or "privacy preservation" or "privacy-preserving XYZ". The title could, for example, be "... privacy-preserving violence detection" or so
* the statement that surveillance actually helps reducing violence is not substantiated with sources- even though ther are tons of literature on this aspect
* "requires obtaining explicit consent" is not true. Under any legislation, there are further modes for legitimizing dat gathering.
* line 53: "avoid data bias" --> which bias?
* related work is a bit short
* line 172/173: references to other DVS datasets definitely needed
* staggering --> other word

**Correctness:**

Most claims seem to be correct, except those referring to representativeness of the dataset etc (see above)

**Documentation:**

yes

**Ethics:**

yes, light concerns. Not for the submission itself but for the line of argumentation it nurtures and fosters. Stating that such data does not pose privacy risks is just too simplistic - see above. For instance, identification is not limited to faces.

**Limitations:**

Unfortunately, it also seems as if the authors did not properly account for ethical, privacy, and societal aspects. Basically, large parts of the respective "argumentation" in the paper as well as in question answers in the the supplementary material boil down to "DVS footage is privacy-preserving in and by itself, given that we don't see faces." For instance, the answer to question A.2.15 more or less negates the identifiability of natural persons in the dataset. The same argument can also be found around lines 35-40. This is by far not sufficient and not in line with long-known research on identifying individuals based on movement (e.g. gait) patterns.

The same rather un- or only superficially reflected tenor also applies to aspects such as the potential negative societal impacts (e.g., checklist question 1c). The question is whether the authors discussed any potential societal impacts of their work. We are in the domain of video surveillance and automated behavior detection. And the authors **just go with a N/A answer?** Im mean... there is a reason why these questions are asked. Different from the statement in the checklist (2b), the limitations **of the presented dataset/approch** are also not discussed - not even superficially. Similarly, I also have my doubts regarding the consent gathered from the students. In the supplementary material, the authors state that consent was collected, but we do not know whether the consenting students were actually aware of the potential risks and impacts of consenting. It is also not 100% clear whether they only consented to being recorded or actually also to having the resulting data mad publicly available. Long story short: Given various warning points, I am not convinced that the authors actiually did their homework in matters of ethics, privacy and societal aspects but rather took respective procedures overly easy.

**Opportunities For Improvement:**

Actual relevance of the dataset is limited in matters of representativeness, diversity and validation:
- Basically, the dataset was created by students *performing* the actions to be included in the dataset. It is obvious that such performed actions do not perfectly resemble actual violent and nonviolent actions. For instance, person A punching person B will lead to very specific movements on B's side in reality, which will presumably not be present in the artificial dataset. It is not sufficiently ensured that these deviations of artificial from actual (esp. violent) actions do not impair a proper assessment of those classifiers the dataset is targeted at. The dataset might be ok for some early-stage research, but for accepting it as an at least implicit reference dataset, my respective uncertainties are too high.
- In addition, the actions to be performed included 6 violent actions and 4 non-violent ones. Given that the dataset is meant to serve as a baseline for distinguishing violent from nonviolent actions, I feel especially the nonviolent ones to be largely underrepresented. Activities (and, thus, respective footage) that should not be classified as violent ones are far more diverse than just "handshaking, finger guessing, greeting, and walking" - what about standing, passing by, lively speaking and gesticulating, speaking on the phone, hugging, kissing, ...? With only these four non-violent options being included, I see a significant risk of misclassification. Similarly, it is also not clear how diverse the dataset is in matters of "actors". Different sizes, wights, genders, and respective ratios between the perons involved might make a significant difference here.
- The dataset was also not properly cross-validated against actual, reliable ground-truth or so to detect and avoid such misclassifications. "Awarding a NeurIPS reference dataset" label under these conditions bears - especially in the context of video surveillance - a particular risk of such misclassifications to persist in(to) practice, with expectable detrimental implications.

**Relation To Prior Work:**

a bit short but ok

**Summary And Contributions:**

This paper presents a dataset of DVS data representing recordings of violent and non-violent behavior. The underlying recordings were taken in a controlled environment with student "actors" playing a variance of 6 different violent and 4 different non-violent behaviors. The dataset is meant as ground-truth baseline for evaluating violence-detecting classifiers particularly tailored to the nature and characteristics of DVS data. The dataset comprises 10,000 scenes ("event segments") and is, to provide ground truth for later automated detection, annotated with human body keypoints (elbows, ...) using a semi-automated procedure applied to traditional video streams corresponding to the DVS segments. In addition to the dataset itself, three classification tasks are proposed as benchmarks to be applied on the dataset: action recognition, temporal action localization and pose estimation. For each of these, reasonable evaluation metrics are proposed and a couple of pre-existing approaches are applied.

---

> ### Author Response · Authors · 2023-08-21
> **Response to reviewer dVaU (1/2)**
>
> Thank you for your detailed and thoughtful feedback. We have addressed each of the concerns you raised and have incorporated the relevant changes in the revised version of the paper.
>
> ### Reply to Opportunities For Improvement:
> Thank you for your insightful comments. Constructing a action recognition dataset by instructing participants to perform specific actions is a prevalent approach, with several datasets developed using similar methods already published [1][2][3][4]. We agree that these datasets, constructed using this method, might slightly deviate from real-life action scenarios. However, designing a dataset intended for robust model evaluation necessitates providing performance metrics that are computationally accessible with minimal influences independent of tasks. We have specifically designed distinct action categories and clear labels to facilitate comparable model training.
> On the other hand, we further clarify that the dataset we constructed is not solely intended for privacy protection or violence detection. Given the current state of violence detection, there is limited research that simultaneously considers privacy concerns, which still a gap in this area. Ordinary and harmless activities are indiscriminately recorded in this state. Therefore, we designed this dataset and established a benchmark for addressing this concern. In the long run, our goal is to have this dataset support violence detection while offering a higher level of privacy protection.
> the primary objective of our dataset is to create one that uses DVS for privacy protection, introducing a novel possibility for ensuring privacy. Given that no dataset had previously been built using this method, our aim is to explore the construction of a violence detection dataset using DVS for privacy protection. Such an approach ensures privacy at the data generation level, rather than pre-processing after acquiring the raw data. Moreover, the variety of actions that individuals perform daily is vast, and attempting to cover all types of actions within a single dataset is challenging. We've selected some of the most common actions for filming for representativeness, which encompass both broad and subtle movements of the hands and feet.
>
> To enhance the clarity and rigor of our paper, we've incorporated a limitation in Discussion of the revised version to address related concerns. We've also added more descriptions concerning data diversity.
> We've incorporated comparative experiments between RGB data and other privacy-protection action recognition methods to demonstrate the performance of data captured by DVS cameras.
>
> ### Reply to Limitations:
> Thank you for your comments and providing us the chance to clarify. We have expanded upon the definition and implications of privacy in our paper. Privacy refers to personal information that an individual does not wish to be accessed or understood by others. The disclosure of such information might result in a loss of benefits or harm to the individual's interests. Privacy preserving involves measures to ensure that private information contained within data is either eliminated or made extremely difficult to extract[5]. Typically, privacy information captured from surveillance images is varied and encompasses aspects such as facial data, gait patterns, and even individuals' habits of daily life. Therefore, in this article, the primary focus of privacy protection is on facial information used in critical payment and unlocking scenarios, rather than broadly on whether an individual can be identified. We regret not explicitly specifying our definition on privacy and its scope in the original paper. Many machine learning-based privacy protection techniques also focus on such facial details[6][7][8]. We've addressed this in the revised manuscript. On another note, we acknowledge the existence of machine learning techniques analyzing motion patterns for human recognition. We've adjusted and provided clarifications on the issue A.2.15 to enhance the rigor of our paper.

---

> > ### Author Response · Authors · 2023-08-21
> > **Response to reviewer dVaU (2/2)**
> >
> > Regarding the societal impact of our research, we've detailed this in our revised manuscript. We apologize for not delving deeper into the potential societal ramifications initially. We've rectified and expanded upon this aspect in our response. Furthermore, we've delineated the limitations of our work to better convey its comprehensiveness. For example, in our paper, it's challenging to cover every possible violent or non-violent action a person might exhibit comprehensively. An incomplete category set might increase the risk of classification errors in the model, which could adversely affect judgments regarding violence.
> > Concerning participant consent, we informed every participant in detail about the filming process and explicitly outlined the content, purpose, and potential risks associated with the data capture. We've appended the instruction manual used during filming in the supplementary material, signifying that participants were provided with comprehensive details. We deeply appreciate your focus on the societal aspects of privacy. We regret to the initial shortcomings in our description, and we've made efforts to provide a more comprehensive account in the revised version.
> >
> >
> > ### Reply to Relation To Prior Work:
> > Thank you for your comment. We have incorporated a description of other action recognition privacy protection methods in the 'Related Work' section of the revised version.
> >
> >
> > Reply to Additional Feedback:
> > 1.Thank you for pointing out that. We have changed our title to Bullying10K: A Large-Scale Neuromorphic Dataset towards Privacy-Preserving Bullying Recognition.
> > 2.We have added the literature to support the statement.
> > 3.In this sentence, we refer to data collection for publication and public dataset. We are sorry for the confusion caused by that statement. We change it to Nonetheless, this data-gathering method frequently requires obtaining explicit consent from recorded participants for public data collection.
> > 4.The bias mentioned here refers to the difference between directly capturing with a DVS camera and using a conversion algorithm.
> > 5.We have incorporated a new description of other action recognition privacy protection methods in the 'Related Work' section of the revised version.
> > 6.We have incorporated related citation.
> > 7.We have changed staggering to an amount of.
> >
> >
> >
> > Thank you again for providing a constructive comment. We hope we have addressed all your questions and concerns. Please let us know if you have any further questions.
> >
> >
> > [1] Blunsden S, Fisher R B. The BEHAVE video dataset: ground truthed video for multi-person behavior classification[J]. Annals of the BMVA, 2010, 4(1-12): 4.
> >
> > [2] Yun K, Honorio J, Chattopadhyay D, et al. Two-person interaction detection using body-pose features and multiple instance learning[C]//2012 IEEE computer society conference on computer vision and pattern recognition workshops. IEEE, 2012: 28-35.
> >
> > [3] Schuldt C, Laptev I, Caputo B. Recognizing human actions: a local SVM approach[C]//Proceedings of the 17th International Conference on Pattern Recognition, 2004. ICPR 2004. IEEE, 2004, 3: 32-36.
> >
> > [4] Shahroudy A, Liu J, Ng T T, et al. Ntu rgb+ d: A large scale dataset for 3d human activity analysis[C]//Proceedings of the IEEE conference on computer vision and pattern recognition. 2016: 1010-1019.
> >
> > [5] Shokri R, Shmatikov V. Privacy-preserving deep learning[C]//Proceedings of the 22nd ACM SIGSAC conference on computer and communications security. 2015: 1310-1321.
> >
> > [6] A survey on gait recognition[J]. ACM Computing Surveys (CSUR), 2018, 51(5): 1-35.
> >
> > [7] Wang Z, Jiang K, Hou Y, et al. A survey on human behavior recognition using channel state information[J]. Ieee Access, 2019, 7: 155986-156024.
> >
> > [8] Marsico M D, Mecca A. A survey on gait recognition via wearable sensors[J]. ACM Computing Surveys (CSUR), 2019, 52(4): 1-39.

---

> > > ### Comment · Reviewer_dVaU · 2023-08-28
> > >
> > > Thank you for taking into account the remarks. I raised the rating by one step (without changing all the text, title etc.). Still not 100% convinced in several regards, but the changes definitely increased the paper's quality.

---

> > > > ### Comment · Reviewer_dVaU · 2023-08-28
> > > > **sry, was on vacation till yesterday.**
> > > >
> > > > ... now responded and slightly changed my rating

---

### Official Review · Reviewer_W6Gq · 2023-07-20
**Bullying10K balances violence detection and personal privacy.**

**Rating:** 7
**Confidence:** 4
**Correctness:** Yes
**Clarity:** Yes, the paper is well written.

**Strengths:**

Using cameras in public spaces to detect and prevent violence is beneficial, but privacy concerns arise. The Bullying10K dataset introduces a novel approach by utilizing Dynamic Vision Sensors (DVS) to capture pixel brightness variations instead of imagery, addressing privacy issues. This innovative concept opens up new possibilities as previous DVS implementations were limited in scale and simplistic labels. With the Bullying10K dataset, complex and rapid actions can be effectively encompassed.

Experimental results on action recognition, temporal action localization, and pose estimation demonstrate the efficacy of Bullying10K in these tasks. It provides a valuable opportunity to develop more advanced algorithms for these specific areas.

**Additional Feedback:**

None

**Documentation:**

Yes

**Limitations:**

See "Opportunities for Improvement", there's a need to discuss a concrete definition of privacy.

**Opportunities For Improvement:**

One notable weakness of the paper is the lack of a clear and detailed definition of privacy preservation. While the paper claims that the use of pixel brightness variations instead of static imagery helps protect privacy, it does not thoroughly address potential concerns. It briefly mentions that there are techniques to reconstruct images from DVS data, albeit with a low success rate. However, the possibility of private information leakage remains a valid concern.

To strengthen the paper's argument and enhance its credibility, it is crucial to provide a more comprehensive and explicit discussion on what it means to be privacy-preserving. This would involve addressing potential vulnerabilities and outlining measures taken to mitigate privacy risks. A detailed examination of the limitations and safeguards implemented within the Bullying10K system would provide a more convincing argument for its privacy-preserving capabilities.

**Relation To Prior Work:**

Yes, there's a comparison.

**Summary And Contributions:**

This paper presents Bullying10K, which balances violence detection and personal privacy. It utilizes Dynamic Vision Sensors (DVS) cameras to detect violent incidents by capturing pixel brightness variations instead of static imagery, thereby preserving privacy.

---

> ### Author Response · Authors · 2023-08-21
> **Response to reviewer W6Gq**
>
> Thank you for your careful and thoughtful comment. We are grateful for your positive feedback on our paper. Below are our detailed comments regarding your questions:
>
> We agree that there should be a clear and detailed definition for privacy persevering. We have discussed more detail in the revised version of our paper to enhance its rigor and make it convincing.
>
> Privacy refers to personal information that an individual does not wish to be accessed or understood by others. The disclosure of such information might result in a loss of benefits or harm to the individual's interests. Privacy preserving involves measures to ensure that private information contained within data is either eliminated or made extremely difficult to extract [1]. Privacy leakage can occur by attacking servers hosting raw data or by launching privacy attacks against models trained using raw data [2]. Typically, privacy information captured from surveillance images is varied and encompasses aspects such as facial data, gait patterns, and even individuals' habits of daily life. Among these, facial information stands out as the most commonly utilized type. It has been widely adopted for applications such as facial payment systems [3], device unlocking [4] , making it one of the most concerned about in terms of privacy.  Therefore, in this article, the primary focus of privacy protection is on facial information used in critical payment and unlocking scenarios, rather than broadly on whether an individual can be identified.
> Our emphasis lies in ensuring that certain privacy protection measures pose substantial barriers to applications that necessitate detailed facial information for accurate identification. Many machine learning-based privacy protection techniques also focus on such facial details [5][6][7].
>
> The dataset we constructed is not solely intended for privacy protection or violence detection. Given the current state of violence detection, there is limited research that simultaneously considers privacy concerns, which still a gap in this area. Our goal is to have this dataset support violence detection while offering a level of privacy protection.
> Furthermore, DVS data protect privacy at the data generation level, distinct from machine learning-based techniques which first produce raw data before undergoing privacy-oriented processing. The leakage of raw data still poses privacy challenges, as attackers can directly access the raw information via camera attack [8]. Additionally, pre-processing of DVS data typically is with less complexity than machine learning-based techniques.
>
> We have incorporated a description of other privacy protection methods pertaining to action recognition in the revised version in the 'Related Work' section. Additionally, we have compared the performance between original RGB images, DVS data, and other privacy protection methods to indicate the utility of our dataset. We have also appended a limitations section (Section 5) to better delineate the dataset's characteristics and the extent of its privacy protection capabilities."
>
> Thank you again for providing a thoughtful comment. We hope we have resolved all your questions and concerns. Please let us know if you have any further questions.
>
>
> [1] Shokri R, Shmatikov V. Privacy-preserving deep learning[C]//Proceedings of the 22nd ACM SIGSAC conference on computer and communications security. 2015: 1310-1321.
>
> [2] Rigaki M, Garcia S. A survey of privacy attacks in machine learning[J]. arXiv preprint arXiv:2007.07646, 2020.
>
> [3]Liu Y, Yan W, Hu B. Resistance to facial recognition payment in China: The influence of privacy-related factors[J]. Telecommunications Policy, 2021, 45(5): 102155.
>
> [4] Vamsi T K, Sai K C, Vijayalakshmi M. Face recognition based door unlocking system using Raspberry Pi[J]. International Journal of Advanced Research, Ideas and Innovation in Technology, 2019.
>
> [5] A survey on gait recognition[J]. ACM Computing Surveys (CSUR), 2018, 51(5): 1-35.
>
> [6] Wang Z, Jiang K, Hou Y, et al. A survey on human behavior recognition using channel state information[J]. Ieee Access, 2019, 7: 155986-156024.
>
> [7] Marsico M D, Mecca A. A survey on gait recognition via wearable sensors[J]. ACM Computing Surveys (CSUR), 2019, 52(4): 1-39.
>
> [8] Ren Z, Lee Y J, Ryoo M S. Learning to anonymize faces for privacy preserving action detection[C]//Proceedings of the european conference on computer vision (ECCV). 2018: 620-636.

---

> > ### Comment · Reviewer_W6Gq · 2023-08-23
> >
> > Thank you for addressing my concerns, I've raised the score.

---

### Official Review · Reviewer_MGZy · 2023-07-21
**Review on the dataset paper Bullying10K**

**Rating:** 5
**Confidence:** 4
**Correctness:** maybe yes. Please find the Limitation…
**Clarity:** Yes.

**Strengths:**

- As asserted by the authors, neuromorphic dataset-based action recognition can be considered as an alternative capable of recognizing actions without compromising privacy.
- The structure of the paper is relatively well-organized, and there were no issues in comprehending its content.
- Performance evaluation was conducted on various models and backbones.


**Additional Feedback:**

There is no additional feedback.

**Documentation:**

Yes.

**Ethics:**

No. There is no ethical problem.


**Limitations:**

- It raises doubts about the practicality of such machine vision techniques in terms of privacy-preservation and cost-effectiveness. The development of privacy-preserving machine vision has been ongoing for several years, with a substantial body of literature available. However, this study fails to address or discuss any of these existing approaches.
- In fact, privacy-preserving vision methods encompass various approaches such as using blurred images, low-resolution images, and radar images. Recently, multiple approaches employing privacy-preserving lenses have been proposed to protect privacy while recognizing pose and action. To substantiate the key claims made in this paper, it is crucial to qualitatively and quantitatively compare the advantages and disadvantages of DVS-based action recognition in contrast to these various privacy-preserving machine vision techniques. When considering real-world deployment scenarios, the necessity for multiple camera installations and preprocessing makes the DVS-based approach appear cost-inefficient compared to other privacy-preserving machine vision technologies.
- DVS-based computer vision was developed to address the issues of data redundancy, high latency, and low temporal resolution associated with traditional RGB image-based computer vision techniques. There is concern that the main argument put forth in this study is merely a reinterpretation, suggesting that DVS-based datasets are also superior in terms of privacy preservation.
- The publicly available dataset can be downloaded from the website and has a total size of approximately 46GB. It is unclear why the paper consistently refers to a dataset size of 100GB.

**Opportunities For Improvement:**

For the core argument presented in this study to be compelling, it is imperative to quantitatively compare the performance of action recognition against other privacy-preserving action recognition techniques.




**Relation To Prior Work:**

Maybe yes, but there is room for improvements. Please find the Limitations section.

**Summary And Contributions:**

This study aimed to construct a neuromorphic dataset using Dynamic Vision Sensors (DVS) cameras for privacy-preserving human action recognition. Subsequently, the constructed dataset was utilized to evaluate the performance of action recognition in relatively complex scenarios.

---

> ### Author Response · Authors · 2023-08-21
> **Respones to reviewer MGZy (1/2)**
>
> We would like to thank you for your responsible comment. Below are our detailed comments regarding your questions:
>
>
>
> ### Reply to Opportunities For Improvement:
> Thank you for your constructive feedback. As a dataset involved to privacy protection, we agree that comparing its performance with other privacy-preserving action recognition techniques is essential. We further clarify that the dataset we constructed is not solely intended for privacy protection or violence detection. Given the current state of violence detection, there is limited research that simultaneously considers privacy concerns. Ordinary and harmless activities are indiscriminately recorded in this state. Therefore, we designed this dataset and established a benchmark for addressing this concern. In the long run, our goal is to have this dataset support violence detection while offering a higher level of privacy protection.
> Accordingly, we've added a section comparing the performance of other privacy-preserving action recognition techniques in Section 4.1. We implemented the DS method described in [1], and the GB method in [2] ,and the BDQ method in [3] on the corresponding RGB frames of the Bullying10k dataset. We observed that after employing privacy protection with RGB data, there is a decline in performance. However, DVS data demonstrated slightly superior results compared to RGB data.  DVS captures event data by detecting changes in brightness. This method inherently offers advantages such as resilience to significant illumination changes, motion blur resistance, and reduced data redundancy. Therefore, for tasks like action recognition that rely on motion characteristics, the features captured by DVS are particularly beneficial. Similar observations and the advantages of DVS can be found in other studies [7][8][9].
>
> | Model                                          | RGB   | DVS   | DS-2[1] | DS-4[1] | GB-3[2] | GB-5[2] | BDQ-1[3] | BDQ-2[3] |
> |------------------------------------------------|-------|-------|----------|----------|----------|----------|-----------|-----------|
> | R3D [4]                                       | 64.00 | 66.80 | 63.30    | 63.15    | 62.70    | 61.45    | 60.10     | 59.75     |
> | SlowFast [5]                                  | 59.25 | 69.00 | 57.80    | 55.40    | 57.20    | 54.95    | 60.20     | 59.45     |
> | X3D [6]                                       | 63.20 | 70.80 | 60.75    | 52.25    | 58.20    | 47.80    | 67.15     | 65.60     |
>
>
>
> ### Reply to Limitations:
> Thank you for your comments. As mentioned in "Reply to Opportunities For Improvement", we have discussed other existing algorithms for privacy preservation. We have added a section (Section 4.1) dedicated to comparing the performance of privacy-preserving action recognition techniques. In addition, discussions on other privacy-preserving action recognition have been incorporated into the related work.
>
> On another note, DVS data protect privacy at the data generation level, distinct from machine learning-based techniques which first produce raw data before undergoing privacy-oriented processing. The leakage of raw data still poses privacy challenges, as attackers can directly access the raw information via camera attack[10]. In addition,  pre-processing of DVS data typically is with less complexity than machine learning-based  techniques.. On the other hand, the storage capacity of DVS data and its transmission efficiency offer substantial advantages and cost benefits compared to traditional RGB data. Using data from a single DVS camera can fulfill the purpose of action recognition.

---

> > ### Author Response · Authors · 2023-08-21
> > **Respones to reviewer MGZy (2/2)**
> >
> > We further clarify that the dataset we constructed is not solely intended for privacy protection or violence detection. Given the current state of violence detection, there is limited research that simultaneously considers privacy concerns. Ordinary and harmless activities are indiscriminately recorded in this state. Therefore, we designed this dataset and established a benchmark for addressing this corcern. In the long run, our goal is to have this dataset support violence detection while offering a higher level of privacy protection.
> >
> > The dataset we constructed is not solely intended for privacy protection or violence detection.Given the current state of violence detection, there is limited research that simultaneously considers privacy concerns, which still a gap in this area. The primary focus of this paper is the creation and capturing of a DVS dataset. Our goal is to have this dataset support violence detection while offering a level of privacy protection, rather than merely reiterating and explaining the characteristics of DVS. We also provide a benchmark for the dataset to measure its performance and compare various models.
> >
> > Notably, our intention is not to negate existing privacy-preserving machine learning vision technologies. Instead, we present this approach offers a novel possibility for privacy preservation, distinct from traditional image surveillance methods. It has a  influence in the field of privacy protection and security surveillance. We will mention further in the revised paper.
> >
> > The size of the dataset after recording is as reported in the paper. For ease of use, we converted it into numpy format and compressed it when we upload, then achieve the size available on the website.
> >
> >
> > Thank you once again for your comment on our paper. We hope our responses have addressed your questions and concerns.
> >
> > [1] Wu Z, Wang H, Wang Z, et al. Privacy-preserving deep action recognition: An adversarial learning framework and a new dataset[J]. IEEE Transactions on Pattern Analysis and Machine Intelligence, 2020, 44(4): 2126-2139.
> >
> > [2] i M, Liu J, Fan H, et al. STPrivacy: Spatio-Temporal Tubelet Sparsification and Anonymization for Privacy-preserving Action Recognition[J]. arXiv preprint arXiv:2301.03046, 2023.
> >
> > [3] umawat S, Nagahara H. Privacy-Preserving Action Recognition via Motion Difference Quantization[C]//European Conference on Computer Vision. Cham: Springer Nature Switzerland, 2022: 518-534.
> >
> > [4] ran D, Ray J, Shou Z, et al. Convnet architecture search for spatiotemporal feature learning[J]. arXiv preprint arXiv:1708.05038, 2017.
> >
> > [5]  eichtenhofer C, Fan H, Malik J, et al. Slowfast networks for video recognition[C]Proceedings of the IEEE/CVF international conference on computer vision. 2019: 6202-6211.
> >
> > [6]  eichtenhofer C. X3d: Expanding architectures for efficient video recognition[C]Proceedings of the IEEE/CVF conference on computer vision and pattern recognition. 2020: 203-213.
> >
> > [7] Tang C, Wang X, Huang J, et al. Revisiting Color-Event based Tracking: A Unified Network, Dataset, and Metric[J]. arXiv preprint arXiv:2211.11010, 2022.
> >
> > [8] Gallego G, Delbrück T, Orchard G, et al. Event-based vision: A survey[J]. IEEE transactions on pattern analysis and machine intelligence, 2020, 44(1): 154-180.
> >
> > [9] Alonso I, Murillo A C. EV-SegNet: Semantic segmentation for event-based cameras[C]//Proceedings of the IEEE/CVF Conference on Computer Vision and Pattern Recognition Workshops. 2019: 0-0.
> >
> > [10] Ren Z, Lee Y J, Ryoo M S. Learning to anonymize faces for privacy preserving action detection[C]//Proceedings of the european conference on computer vision (ECCV). 2018: 620-636.

---

> ### Author Response · Authors · 2023-08-30
> **Response to reviewer**
>
> Dear reviewer,
>
> We greatly appreciate your valuable comment once again, and we have already made modification based on your suggestions. As the discussions phase are nearing conclusion, please feel free to inform us if you have any further suggestions or questions.
>
> Best,
>
> Authors

---

### Official Review · Reviewer_4C71 · 2023-08-02
**Review of NeurIPS 2023 Track Datasets and Benchmarks Submission97**

**Rating:** 5
**Confidence:** 3

**Strengths:**

The dataset provided in this paper is designed to address the privacy issues in the training of sensitive action detection models, and the dataset itself has the advantage of being desensitized, which is larger and more complex than the existing dataset and provides data support for the violent action recognition system based on the desensitized dataset. It has a positive social influence in the field of privacy protection and security surveillance.

**Additional Feedback:**

No additional feedback

**Clarity:**

The paper is well-written in general, but there are some shortcomings in the details.
1.	Formatting and formality: The description of figures shall be improved, including punctuation and caption. The words in Figure 3 overlapped with the image
2.	Insufficiently comprehensive references to benchmarking methodologies and lack of clarity and notability in the mathematical presentation of metrics.


**Correctness:**

The paper as a whole constructs a privacy-insensitive dataset that can be relied upon to build a viable violent action detection system. However, the following issues require further discussion:
1.	Is the process of constructing the dataset fully privacy-preserving? Can the model locate individuals based on their actions alone?
2.	Does DVS data give sufficient performance guarantees for target detection tasks compared to image data? This should be presented in the dataset and benchmark


**Documentation:**

The article provides easy-to-use Pytorch-oriented interfaces and download channels for the dataset.
However, there is no material or tools provided for the recommended benchmark and train/val partitioning of the dataset. This makes the benchmark in this paper difficult to reproduce


**Ethics:**

Since the dataset involves violent actions, the article does not mention whether the data collection was adequately informed and consented to by the participants

**Limitations:**

I could not find the corresponding statements about the limitations of the proposed Bullying 10K. I do think that the authors should provide discussions about the potential societal impact and the limitations of this work itself.

**Opportunities For Improvement:**

Details about the data collection process, like camera setup, capture protocol etc. are unclear.
The diversity and complexity of the dataset is not fully discussed, including the camera position, composition of participants and environmental factors.
The sensitivity of DVS cameras is not adequately discussed.
The article doesn't mention how the training, validation and test sets are divided. This does not seem to make sense.

**Relation To Prior Work:**

Yes

**Summary And Contributions:**

The paper introduces a new large-scale event-based dataset called Bullying10K for detecting violent behavior.
Three benchmarks, including an action recognition benchmark, an action temporal localization benchmark, and a pose estimation is provided.
The goal is to enable training systems to detect violence while respecting privacy. The dataset is captured using Dynamic Vision Sensor (DVS) cameras, which record changes in pixel brightness instead of static images and is practical in privacy-preserving scenarios.
The key contribution is introducing a large-scale, privacy-focused event-driven dataset capturing complex violent behaviors, which facilitates research on violence detection and privacy-preserving video systems.

---

> ### Author Response · Authors · 2023-08-21
> **Respones to reviewer 4C71 (1/3)**
>
> We thank the reviewer for their valuable time and constructive feedback on this paper. Below, we have provided detailed responses to each of your comments.
>
> ### Reply to Opportunities For Improvement:
>
>
> Thank you for your responsible feedback. We have added more details regarding the data collection. We used two Davis346 cameras for data acquisition. To capture multiple viewing angles and ensure diversity in the collected data, we position two DVS cameras on the left and right sides of the filming scene, as depicted in Figure 2. For consistency in the dataset, the cameras were positioned 5 meters apart, with both camera lenses were oriented at a 30-degree angle from the direct front. Additionally, To capture a diverse range of data and to more closely align with real-world conditions, we set up two lighting conditions: light and dark.  The sensitivity of the DVS cameras allows for a time precision of less than 1us, which results in more noise under dark conditions compared to light ones. The camera lenses have a focal length of 4mm, an aperture of 1.6, and an exposure time of 20ms to ensure an appropriate filming range and exposure.We invited 25 distinct participants, leading to 50 recording groups in total. The gender ratio among participants was 1:1. Instead of merely repeating a specific action, participants were encouraged to freely execute movements, while ensuring the action type remained consistent, adding to the dataset's diversity and complexity.
>
> Regarding data partitioning, we've mentioned our dataset's division method for each task setting in revised paper. For the action recognition task, The Bullying10K dataset has been divided into training and validation sets with an 8:2 ratio. We provided a corresponding interface in the code to ensure data consistency. For the temporal action localization task, To achieve more accurate testing results, we provided action recognition data for the pre-trained features at the same ratio. Additionally, we trained the model for the localization task at a 1:3 ratio. Similarly, for the pose estimation task, we split the dataset at an 8:2 ratio and have provided the corresponding pose annotations along with the data to ensure consistency.
>
> ### Reply to Limitations:
>
> Thank you for pointing that out. We agree that there should be a more comprehensive discussion on social implications and limitations. We have corporated this section in section 5 of the revised paper. We further clarify that the dataset we constructed is not solely intended for privacy protection or violence detection. Given the current state of violence detection, there is limited research that simultaneously considers privacy concerns. Ordinary and harmless activities are indiscriminately recorded in this state. Therefore, we designed this dataset and established a benchmark for addressing this corcern. In the long run, our goal is to have this dataset support violence detection while offering a higher level of privacy protection. Utilizing the properties of the DVS camera, it made privacy protection feasible. This approach offers a novel possibility for privacy preservation, distinct from traditional image surveillance methods. As you mentioned in the strengths, it has a positive societal influence in the field of privacy protection and security surveillance.
> However, this dataset have its limitations. Typically, the kind of private information that can be captured by cameras is varied and encompasses aspects like facial details, gait patterns, and even individuals' habits of daily life. Some recent technologies can identify specific individuals using non-facial information[1][2][3]. Our dataset might struggle to prevent leaks of non-facial information, such as gait data.
> Still, we wish to emphasize that facial data is the most commonly used and has been incorporated into many crucial applications. This type of privacy is our primary focus in this paper. For the reason for clarity in labeling during the dataset creation and for comparability during model validation, our dataset defines specific, commonly observed actions. It's challenging to cover every possible violent or non-violent action a person might exhibit comprehensively.

---

> > ### Author Response · Authors · 2023-08-21
> > **Respones to reviewer 4C71 (2/3)**
> >
> > ### Reply to Correctness:
> >
> > Thank you for your comment. Regarding the first question: we addressed this concern initially in the "Limitations" section of our paper.We agree that there are multiple types of private information present in images, and it's challenging for any dataset to provide comprehensive protection to every detail. In this paper, our focus is on the kind of privacy which, if leaked, could result in potential harm or loss to an individual. We are less concerned with whether or not an individual can be identified. In practical scenarios, facial information draws the most attention. It has become a widely adopted tool for applications, especially those demanding precise recognition such as payments and device unlocks [4][5]. For a thorough treatment of the topic, our paper delves deeper into the definition and scope of privacy, and also discusses the identifiability of other private attributes in the "Limitations" section to ensure the rigor of our work.
> >
> > Regarding the second question: to demonstrate the real-world utility of our dataset, we provided corresponding results in the benchmark section to illustrate that the event data from DVS is sufficient to maintain the performance required for action recognition tasks. Given that both Reviewer 4C71 and Reviewer MGZy concern the performance comparison between privacy methods, we have also included results comparing our method with other privacy-preserving techniques. Our findings indicate that event-based data not only meets the demands of action recognition but, to some extent, enhances performance. DVS captures event data by detecting changes in brightness. This method inherently offers advantages such as resilience to significant illumination changes, motion blur resistance, and reduced data redundancy. Therefore, for tasks like action recognition that rely on motion characteristics, the features captured by DVS are particularly beneficial. Similar observations and the advantages of DVS can be found in other studies [6][7][8].
> > | Model                                          | RGB   | DVS   | DS-2[9] | DS-4[9] | GB-3[10] | GB-5[10] | BDQ-1[11] | BDQ-2[11] |
> > |------------------------------------------------|-------|-------|----------|----------|----------|----------|-----------|-----------|
> > | R3D [12]                                       | 64.00 | 66.80 | 63.30    | 63.15    | 62.70    | 61.45    | 60.10     | 59.75     |
> > | SlowFast [13]                                  | 59.25 | 69.00 | 57.80    | 55.40    | 57.20    | 54.95    | 60.20     | 59.45     |
> > | X3D [14]                                       | 63.20 | 70.80 | 60.75    | 52.25    | 58.20    | 47.80    | 67.15     | 65.60     |
> >
> >
> >
> >
> > ### Reply to Clarity:
> > Thank you for acknowledging the quality of our paper. We have improved the description for each figure and enhanced the captions. Please refer to the updated version of the paper we've submitted. We have also reformatted Figure 3 to make it clearer. Additionally, we have included more details and references about the benchmarking methodology and have mentioned the metrics more clearly. Please check Section 4.3 in the revised paper.
> >
> >
> >
> > ### Reply to Documentation:
> > Thank you for pointing that out. In the revised version of our paper, we have provided a more detailed explanation of the dataset division.  Specifically, we have elaborated on the dataset partitioning in the 'Reply to Opportunities For Improvement' section."
> >
> >
> > ### Reply to Ethics:
> > Thank you for your comment. We agree that obtaining informed consent from participants is crucial when collecting data. In our initial version, we provided a section regarding participant informed consent in the supplementary materials. To further clarify that we have obtained informed consent from all participants, we have added more details in the supplementary materials. Additionally, we have included the information booklet used when collecting the dataset, demonstrating that participants were fully informed about the data collection process.
> >
> > Thank you once again for your feedback on our paper. We hope our responses have addressed your questions and concerns.

---

> > > ### Author Response · Authors · 2023-08-21
> > > **Respones to reviewer 4C71 (3/3)**
> > >
> > > [1] A survey on gait recognition[J]. ACM Computing Surveys (CSUR), 2018, 51(5): 1-35.
> > >
> > > [2] Wang Z, Jiang K, Hou Y, et al. A survey on human behavior recognition using channel state information[J]. Ieee Access, 2019, 7: 155986-156024.
> > >
> > > [3] Marsico M D, Mecca A. A survey on gait recognition via wearable sensors[J]. ACM Computing Surveys (CSUR), 2019, 52(4): 1-39.
> > >
> > > [4] Liu Y, Yan W, Hu B. Resistance to facial recognition payment in China: The influence of privacy-related factors[J]. Telecommunications Policy, 2021, 45(5): 102155.
> > >
> > > [5] Vamsi T K, Sai K C, Vijayalakshmi M. Face recognition based door unlocking system using Raspberry Pi[J]. International Journal of Advanced Research, Ideas and Innovation in Technology, 2019.
> > >
> > > [6] Tang C, Wang X, Huang J, et al. Revisiting Color-Event based Tracking: A Unified Network, Dataset, and Metric[J]. arXiv preprint arXiv:2211.11010, 2022.
> > >
> > > [7] Gallego G, Delbrück T, Orchard G, et al. Event-based vision: A survey[J]. IEEE transactions on pattern analysis and machine intelligence, 2020, 44(1): 154-180.
> > >
> > > [8] Alonso I, Murillo A C. EV-SegNet: Semantic segmentation for event-based cameras[C]//Proceedings of the IEEE/CVF Conference on Computer Vision and Pattern Recognition Workshops. 2019: 0-0.
> > >
> > > [9] Wu Z, Wang H, Wang Z, et al. Privacy-preserving deep action recognition: An adversarial learning framework and a new dataset[J]. IEEE Transactions on Pattern Analysis and Machine Intelligence, 2020, 44(4): 2126-2139.
> > > [10] Li M, Liu J, Fan H, et al. STPrivacy: Spatio-Temporal Tubelet Sparsification and Anonymization for Privacy-preserving Action Recognition[J]. arXiv preprint arXiv:2301.03046, 2023.
> > >
> > > [11] Kumawat S, Nagahara H. Privacy-Preserving Action Recognition via Motion Difference Quantization[C]//European Conference on Computer Vision. Cham: Springer Nature Switzerland, 2022: 518-534.
> > >
> > > [12] Tran D, Ray J, Shou Z, et al. Convnet architecture search for spatiotemporal feature learning[J]. arXiv preprint arXiv:1708.05038, 2017.
> > >
> > > [13] Feichtenhofer C, Fan H, Malik J, et al. Slowfast networks for video recognition[C]Proceedings of the IEEE/CVF international conference on computer vision. 2019: 6202-6211.
> > >
> > > [14] Feichtenhofer C. X3d: Expanding architectures for efficient video recognition[C]Proceedings of the IEEE/CVF conference on computer vision and pattern recognition. 2020: 203-213.

---

> ### Author Response · Authors · 2023-08-30
> **Response to reviewer**
>
> Dear reviewer,
>
> We sincerely appreciate your constructive feedback once again, and we have already implemented changes based on your suggestions. As the discussions phase are nearing conclusion, please feel free to inform us if you have any further suggestions or questions.
>
> Best,
>
> Authors

---

### Author Response · Authors · 2023-08-21
**General response to reviewers**

We appreciate to each reviewer for their valuable time and insightful feedback. The comments from the reviewers have significantly enhanced the rigor and clarity of our paper. In the revised version of the paper, we have made modifications based on the suggestions provided by the reviewers.

This paper introduces a DVS (Dynamic Vision Sensor) based violence recognition dataset that incorporates privacy protection. Given the current state of violence detection, there is limited research that simultaneously considers privacy concerns. Ordinary and harmless activities are indiscriminately recorded in this state. Therefore, we designed this dataset and established a benchmark for addressing this concern. Our goal is to have this dataset support violence detection while offering a higher level of privacy protection. Utilizing the properties of the DVS camera, it made privacy protection feasible. This approach offers a novel possibility for privacy preservation, distinct from traditional image surveillance methods.

* We have included a definition of the scope and meaning of privacy and have expanded on the description of privacy protection.

* We have conducted additional experiments, comparing the performance gaps between various privacy-preserving action recognition algorithms on our dataset. We've also added more descriptions on privacy-preserving algorithms in related work.

* We have incorporated the limitations of our dataset and its potential societal impacts. Furthermore, we've provided more information about the instructions given to participants and their informed consent process.

* We have included More detailed explanations about the dataset creation process, along with a more comprehensive description of the dataset. We have also added further details about the experiments and made modifications to certain sections of the paper.

---

### Author Response · Authors · 2023-08-28
**Response to reviewers**

Dear Reviewers,

We are grateful to the reviewers for investing time in a comprehensive evaluation of our article. We have made revisions based on the reviewer's suggestions. We highly value their input and would be grateful for any further suggestions they might have to enhance our paper.

Best,

Authors

---

### Decision · Program_Chairs · 2023-09-22

**Decision:**

Accept (Poster)

**Comment:**

**Preface.** The average score of this paper is currently 5.75 (7, 6, 5, 5), which is below the acceptance threshold. However, I am taking into consideration the overall quality of the work and the fact that the reviewers with scores 5 (4C71, MGZy) did not return for the rebuttal discussion despite reminders. The authors have comprehensively addressed the comments of these reviewers and incorporated the changes into the revision as I discuss below, and the reviewers have also noted the potential positive impact of this paper, so if the reviewers had even raised their score to a 6, the average score would be over the acceptance threshold. As such, I am consequently considering the paper quality and content of the discussion in my recommendation to the SACs and PCs beyond the numerical score.

**Summary.** This paper proposes Bullying10K, a dataset containing acted violent and non-violent events for supporting violence detection in a privacy-preserving manner. The specific gap that this work addresses is the lack of a dataset that simultaneously considers violence detection *and* privacy preservation.

**Strengths / Reasons to accept.** The reviewers have noted that the paper is structured and well-organized (MGZy), has a positive social influence in the field of privacy protection and security surveillance (4C71), is substantial (dVaU), and provides a valuable opportunity to develop more advanced algorithms for these specific areas (W6Gq).

**Weaknesses / Reasons to reject and author response.** The primary reasons as enlisted by the reviewers are as follows:
- *Details of the capture setup / data collection*. 4C71 pointed out the lack of details on camera setup, protocol, participant composition, and environmental factors. I find that the authors have adequately addressed these in the revised Sec. 3.1.  They have also added the exact informed consent form to the Supplement.
- *Contextualizing the scope of privacy considered*: 4C71, W6Gq, and dVaU raised points about the notion of privacy in this work and whether it is preserved in the data construction. The authors have contextualized this adequately in my opinion, both here and in the revised manuscript. While behavior as biometrics remains a developing research area, the revised contextualization does not diminish the usefulness of this dataset in my opinion.
- *Experiments comparing against other privacy-preserving methods*. 4C71 and MGZy raised concerns about comparing against privacy-preserving techniques for action recognition. The authors have added experiments to revised Sec. 4.1 and the results support the argument for the utility of the DVS data.
- *Societal impact and limited non-violent categories*. dVaU mentioned the lack of discussion of potential negative societal impact and the issue of limited non-violent categories. The authors have acknowledged these and addressed them in the discussion. As it stands, the provided categories still make this a useful dataset for the community.

**Final recommendation.** Given these improvements to the manuscript, I believe the concerns as stated by the reviewers have been addressed and a raised score is warranted. As such, the benefits of the dataset to the community now outweigh the stated concerns. There are remaining issues (see below), which I believe can be addressed in writing and are insufficient for rejection.

**Recommendations to authors.** Authors, while I am recommending accepting this paper, I do so with the caveat of incorporating the following suggestions:

- **Your Datasheet is a bit thin on details.**
  - You've incorporated details about the data collection (response to 4C71) in the main paper, and yet have not mentioned these details (camera details, positions, participants, etc.) in the Collection part of the Datasheet. These exist for a reason, and I suggest you be comprehensive in describing the details.
  - Moreover, given your acknowledgment of dVaU's points of societal impact, I find your Section on Uses to be puzzling. A response of "No" to A.4 and A.5 in light of your discussion of the potential negative societal impact on violence detection is unacceptable. I recommend you update these with a detailed description of potential risks.
  - An ethic's board review should have been done for this paper. I suggest adding a discussion on compliance with China's PIPL.
- **Please re-cite the references** from the rebuttal discussion back in the paper discussion / datasheet. In your response to W6Gq you contextualize the role of faces as a privacy threat with references, and seem to have added these to the introduction. However, your mention of data leakage in the Discussion and A.2.15 is again thin, and could use replicating the discussion on various means of data leakage, including, but not limited to, mentioning the work on gait detection you discuss in the rebuttal.